# CoIDO: Efficient Data Selection for Visual Instruction Tuning via Coupled Importance-Diversity Optimization

**Yichen Yan[1], Ming Zhong[1], Qi Zhu[1], Xiaoling Gu[2], Jinpeng Chen[3], Huan Li[1]***

[1] College of Computer Science and Technology, Zhejiang University
[2] Hangzhou Dianzi University, Hangzhou, China
[3] School of Computer Science (National Pilot Software Engineering School), BUPT
{yichen.yan, chime, lihuan.cs}@zju.edu.cn,
qizhu.zju.research@gmail.com, guxl@hdu.edu.cn, jpchen@bupt.edu.cn

## Abstract

Multimodal large language models (MLLMs) rely heavily on instruction tuning to align vision and language capabilities, yet the computational cost of training on large-scale datasets remains a major bottleneck. Existing data selection methods aim to mitigate this by selecting important and diverse subsets, but they often suffer from two critical drawbacks: high computational overhead from processing the entire dataset and suboptimal data selection due to separate treatment of importance and diversity. We introduce CoIDO, a novel dual-objective framework that jointly optimizes data importance and diversity to overcome these challenges. Unlike existing approaches that require costly evaluations across the whole dataset, CoIDO employs a lightweight plug-in scorer. This scorer is trained on just a small random subset of data to learn the distribution of the candidate set, drastically reducing computational demands. By leveraging a homoscedastic uncertainty-based formulation, CoIDO effectively balances importance and diversity during training, enabling the scorer to infer CoIDO scores for all samples. This unified scoring approach allows for direct ranking and selection of the most valuable subsets, completely avoiding the need for specialized algorithms. In our experiments, we train the CoIDO Scorer using only **20%** of randomly sampled data. Once trained, CoIDO is applied to the entire dataset to select a **20%** subset for instruction tuning. On the widely used LLaVA-1.5-7B model across ten downstream tasks, this selected subset achieves an impressive **98.2%** of the performance of full-data fine-tuning, on average. Moreover, CoIDO outperforms all competitors in terms of both efficiency (lowest training FLOPs) and aggregated accuracy. Our code is available at https://github.com/SuDIS-ZJU/CoIDO.

## 1 Introduction

Instruction tuning has become fundamental for aligning Multimodal Large Language Models (MLLMs) with human intent, empowering models such as GPT-4o [1], Gemini [2], and LLaVA [3] to handle diverse downstream tasks including visual question answering [4–6], image-text retrieval [7–9], and visual grounding [10–12]. While large-scale visual instruction datasets (e.g., LLaVA-665K [13]) have enabled impressive performance, they introduce substantial redundancy, high computational costs, and optimization inefficiencies. For example, fine-tuning a single epoch of LLaVA-1.5-7B typically requires over 20 GPU hours on 8×A100 40GB GPUs [13]. Such computational demands

---

*Corresponding author

39th Conference on Neural Information Processing Systems (NeurIPS 2025).

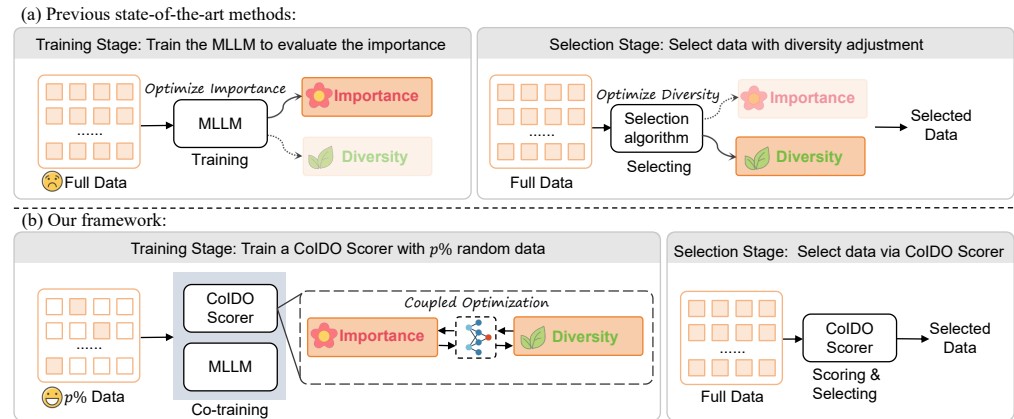

Figure 1: Previous state-of-the-art methods (e.g., TIVE [18] and ICONS [19]) treat importance and diversity as decoupled and independent components, using the entire data in the training stage. Our CoIDO integrates importance and diversity in a coupled and reciprocal optimization, achieving superior data selection by utilizing only a fraction $p\%(p \ll 100)$ of the full dataset for model training and without any specialized algorithm in the selection stage.

pose significant challenges for efficient deployment, particularly for research groups or organizations with limited access to large-scale hardware resources. Recent works, such as LIMA [14], indicate that carefully selecting a small subset of high-quality instructions significantly reduces computational costs while preserving model performance. To mitigate these challenges, data selection methods have been proposed to identify high-quality instruction data that can enhance the performance of large language models (LLMs).

Early data selection methods [14–17] have primarily focused on text-only instruction tuning for LLMs, with limited exploration in the multimodal domain. Unlike text-only data, visual instruction tuning introduces additional complexity due to its bimodal nature. This complexity brings two more challenges for the data selection task: (1) processing image-text data pairs demands greater computational resources, and (2) evaluating the semantic alignment between images and text is inherently more complex. Consequently, it becomes more challenging to account for both the *importance* of individual samples and the *diversity* of the dataset as a whole. As illustrated in Figure 1 (a), current approaches tailored for MLLM [18–20] typically involve two stages: the training stage and the data selection stage. For importance optimization, existing approaches typically require the target MLLM to process the entire candidate dataset in the training stage, either by evaluating importance indicators such as gradients [18, 19], or by leveraging intermediate layer features for clustering [21] to compute transferability, which can be regarded as importance. For diversity optimization, they also need to design a complicated selection algorithm in the selection stage, such as a soft sample [18] or a similarity penalty [20] to optimize the diversity of the subset.

However, these methods have two major drawbacks: (1) The training and selection stages are decoupled processes. Data importance is assessed during training, while diversity is handled separately through specialized algorithms in the selection stage. This separation fails to jointly optimize the two objectives and often results in a suboptimal trade-off. Specifically, placing too much emphasis on importance may reduce overall dataset diversity, while prioritizing diversity could exclude high-value samples. (2) They require the target MLLM training to process the entire dataset to compute sample importance. This results in computational costs comparable to full fine-tuning. This fundamentally ***contradicts*** the objective of data selection, which is to reduce training overhead. Moreover, any addition of new data necessitates reprocessing the entire dataset, severely limiting scalability and efficiency. Therefore, an ideal data selection framework should satisfy two key criteria:

*(i) jointly optimize data importance and diversity*, and    *(ii) significantly reduce computational overhead by utilizing only a limited subset of candidate data before the selection.*

Building on these considerations, we propose a novel and efficient data selection framework, named CoIDO (**Coupled Importance-Diversity Optimization**), explicitly designed to overcome the two critical issues identified above. As illustrated in Figure 1 (b), CoIDO utilizes only $p\%$ (e.g., up

to 20%) of randomly sampled data to train a lightweight scorer, instead of fine-tuning the target MLLM extensively to evaluate the entire dataset. Specifically, in the training stage, we introduce a plug-in COIDO Scorer. Unlike previous methods that require evaluating each data sample during training, COIDO Scorer learns the data distribution from a small subset. Additionally, we move the optimization of data diversity, which was traditionally handled in the selection stage by a selection algorithm, to the training stage. We design an importance loss based on the backpropagation and a diversity loss integrating the spectral clustering. These two objectives are jointly optimized during the training stage via a homoscedastic uncertainty-based optimization [22]. The scorer assigns each data sample a COIDO score that takes both importance and diversity into consideration.

In the selection stage, we select the top-scoring samples from each task's candidate pool in identical proportions. This ensures balanced coverage and prevents task bias that might arise from global ranking and selection. Our method only requires examining a small amount of data, which greatly reduces the training cost. More importantly, since our scorer learns the underlying data distribution efficiently from a small subset, it transfers seamlessly to new in-domain candidate data without retraining, substantially enhancing the scalability in data selection. In summary, our contributions are highlighted in three aspects:

(*i*) We introduce a novel dual-objective optimization approach for MLLM visual instruction data, using coupled optimization for data importance and diversity. (*ii*) We propose a lightweight scorer pipeline that learns the candidate data distribution from a small amount of data in the training stage, enabling efficient and accurate ranking across the entire dataset at significantly reduced training cost. (*iii*) In the selection stage, COIDO Scorer can directly select data from the entire dataset without additional diversity optimization algorithms. Experiments show that fine-tuning the LLaVA-1.5-7b with only 20% of the samples selected by COIDO achieves 98.2% of the performance of using the full dataset.

## 2 Related Work

**Data Selection for LLMs**. Data selection is critical for optimizing both pre-training and fine-tuning of LLMs. For ***pre-training***, several methods [23–26] curate diverse large-scale datasets to improve foundation models, but focus on general-purpose diversity rather than instruction-specific quality. During ***instruction tuning***, where instruction quality is crucial, recent methods like LIMA [14] and ALPAGASUS [15] employ gradients or an LLM-based evaluator to filter low-quality samples. Similarly, NUGGETS [27] utilizes zero-shot and one-shot evaluation, while CAR [28] incorporates expert-aligned scoring with clustering to preserve diversity. Though effective, these methods are designed for *text-only* datasets and fail to account for more complex semantics in multimodal datasets.

Data filtering methods based on static metrics [29–33] are not exclusive to textual data. However, they demonstrate suboptimal performance in the context of visual instruction tuning (see Table 1).

**Data Selection for Visual Instruction Tuning**. MLLMs such as Flamingo [34], LLaVA [3], and BLIP2 [35] rely on instruction tuning to align visual and textual modalities. Early efforts like INSTRUCTIONGPT-4 [36] target small-scale datasets (e.g., MiniGPT-4 [37], 3.4K instructions) and require evaluation across multiple downstream tasks as labels, limiting scalability. More recent methods attempt to address this by prioritizing high-value samples from larger instruction datasets, such as LLaVA-665K [13]. Typical methods include SELF-FILTER [20] that employs a scoring network, TIVE [18] based on gradient to compute data influence, ICONS [19], which applies gradient-based selection to identify representative data from tasks, and COINCIDE [21], which employs clustering techniques based on concept-skill composition representations.

These methods [20, 18, 19, 21] typically require full-dataset processing by the target MLLM, incurring high computational cost. They also decouple the optimization of importance and diversity, often leading to suboptimal subsets. In contrast, our proposed COIDO reduces overhead by training a lightweight scorer on a small subset, while jointly optimizing importance and diversity. Moreover, our approach can transfer seamlessly to new in-domain candidate data without retraining

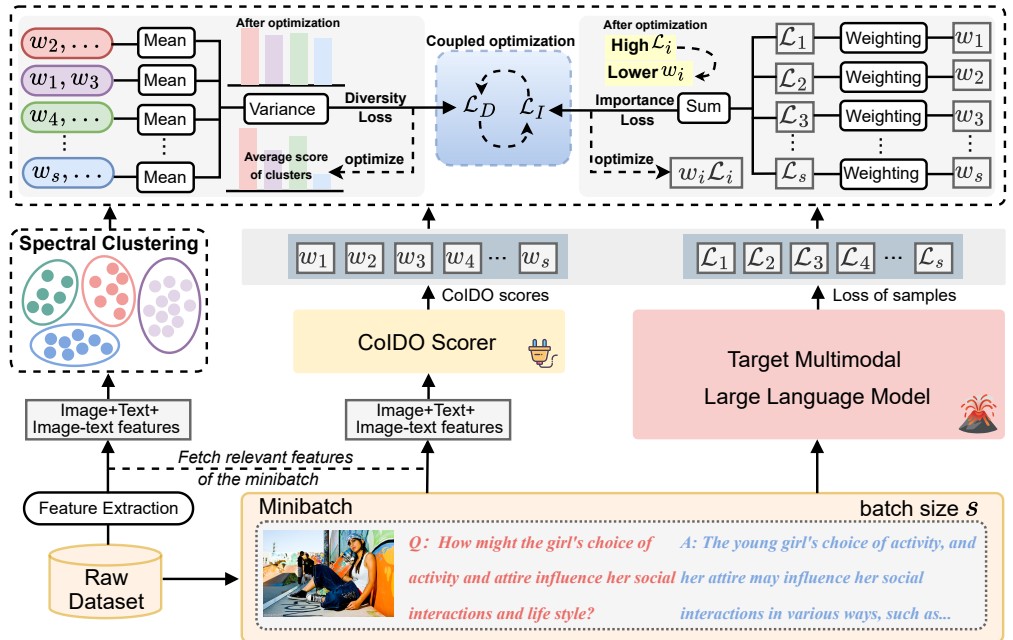

Figure 2: Framework overview of CoIDO (also see Appendix A.2). Basic features (image, text, and image-text) are extracted for spectral clustering and scorer training. The plug-in CoIDO Scorer, co-trained with the target MLLM (e.g., LLaVA-1.5-7B), outputs a score $w_i$ for each sample. These scores are used to compute the importance loss $\mathcal{L}_I$ and diversity loss $\mathcal{L}_D$, which are jointly optimized to capture both data importance and diversity via backpropagation using batches of random samples.

## 3 Methodology

### 3.1 Task Description, Framework, and Key Modules

**Task Description**. Given a large-scale visual instruction dataset $\mathcal{D} = \{z_j\}_{j=1}^N$, where each sample $z_j = (x_j, y_j)$ includes a visual instruction $x_j$ captured by an image-text pair (see Figure 2 bottom-right), and a target response $y_j$, the task of ***data selection for visual instruction tuning*** is to efficiently select a subset $\mathcal{D}_h \subset \mathcal{D}$ with $|\mathcal{D}_h| = \gamma N$. The subset should reduce visual instruction tuning costs while maintaining at least *Rel.* percentage of the performance achieved using the full dataset. Typically, $\gamma \in [0.1, 0.4]$ and *Rel.* $\geq 95\%$ are desired by real-world applications [20, 18, 19, 21].

**Overall Framework**. Our proposed CoIDO framework efficiently selects high-quality subsets by leveraging only a fraction $p\%$ (e.g., 20%) of random samples for training. As shown in Figure 2, we first extract multimodal features and scoring metrics (evaluating text, images, and image-text alignment) from the raw dataset. These features are used for two purposes: (i) training the lightweight CoIDO Scorer to assess data importance and diversity in one pass, and (ii) clustering to obtain class assignments for each data sample, which will be subsequently used in modeling the diversity loss.

This selection framework, centered around the CoIDO Scorer, is then trained using the sampled data and extracted features in a minibatch mode: basic features are fused and fed into the scorer, while the raw samples are simultaneously used to fine-tune the target MLLM.

During training, the CoIDO Scorer assigns a scalar score $w_i$, referred to as the CoIDO score, to each sample. This score reflects both data importance and diversity due to the coupled optimization of these objectives (see Section 3.2). In the selection phase, the trained scorer is applied to the full dataset to infer the CoIDO score for each sample. The final subset is obtained by ranking the samples based on their scores and selecting the top fraction $\gamma$ per downstream task.

**Feature Extraction**. As depicted in Figure 2, we extract a set of basic features for samples to train the scorer and perform clustering. These include: 1) *Text Features*: Captured using an LLM-based evaluator, which assesses dimensions like spelling and grammar quality. The resulting score,

referred to as the *LLM Score*, is detailed in Appendix A.1. 2) *Image Features*: Measured using the *ImageReward Score* [38], which evaluates image fidelity and semantic clarity. 3) *Image-Text Features*: Derived from a pre-trained CLIP encoder [39], including the *low-dimensional multimodal representations* of image-text pairs and the corresponding *CLIP Score* [29] on image-text alignment. These features collectively provide a comprehensive assessment of text quality, image quality, and image-text alignment. The combined features are concatenated as input for clustering and training.

**Spectral Clustering**. We use *spectral clustering* [40] to group data samples into a total of $M$ classes. Unlike $K$-means, spectral clustering is better suited for capturing complex, non-linear distributions in multimodal data. The resulting clusters represent fundamental class information, which is used to compute diversity loss during optimization. Further implementation details and experiments on feature usage and clustering methods are provided in Appendixes A.1, C.1, and C.2.

**CoIDO Scorer Training**. The scorer is co-trained with the target MLLM to assign CoIDO scores to data samples for the subsequent selection stage, where the features of samples are fetched from the previous clustering stage. During training, we randomly sample a fraction $p\%$ of the dataset. Each batch serves two purposes: (1) fine-tuning the target MLLM using standard instruction learning, and (2) training the plug-in CoIDO Scorer, which processes corresponding basic features via a lightweight architecture, like a multilayer perception (MLP). For each batch, the CoIDO Scorer outputs a CoIDO score per sample, while the target MLLM computes a cross-entropy loss. These outputs are jointly optimized to balance data importance and diversity, as detailed in Section 3.2. We further explore alternative CoIDO Scorer architectures and optimization methods in Section 4.2.

**Importance Loss**. Training loss is a well-established indicator of instruction difficulty, as higher losses typically correspond to more challenging and valuable samples for improving model generalization [41, 42, 20]. Building on this insight, we introduce learnable CoIDO scores, predicted by the scorer, to re-weight the prediction loss for each data sample.

Since more difficult instructions generally result in higher losses, the corresponding learnable scores are expected to decrease during training, thereby implicitly capturing sample importance. For each batch, the CoIDO scores are processed using Softmax and subsequently applied to modulate the cross-entropy (CE) loss of the target MLLM, producing a weighted sum over all samples (see top-right of Figure 2).

Let $s$ represent the number of samples in a batch, drawn from $m$ clusters (where $m \leq M$, as some clusters may not appear in this batch). For the $i$-th cluster, with $n_i$ samples ($s = \sum_{i=1}^{m} n_i$), the importance loss $\mathcal{L}_I$ is defined as:

$$\mathcal{L}_I = \sum_{i=1}^{m} \sum_{k=1}^{n_i} w_{ik} \cdot \text{CE}(y_{ik}, \hat{y}_{ik}), \tag{1}$$

where $y_{ik}$ and $\hat{y}_{ik}$ denote the ground truth and predicted labels for the $k$-th sample in the $i$-th cluster, respectively. We apply a Softmax operation to obtain normalized CoIDO scores $w_{ik}$, which will be used in coupled optimization of the framework (see Section 3.2).

A high cross-entropy $\text{CE}(y_{ik}, \hat{y}_{ik})$ implies the sample is difficult to learn and thus more important. According to the principle of backpropagation, to minimize the overall loss, the model naturally adjusts the weight $w_{ik}$ downward for high-difficulty samples. In other words, a lower $w_{ik}$ corresponds to a more important sample. The formulation $\mathcal{L}_I$ is therefore designed to capture and optimize the importance of individual samples in the data selection process.

**Diversity Loss**. While the importance loss $\mathcal{L}_I$ optimizes sample weights $w_{ik}$ from the data importance perspective, it can lead to *imbalanced* weight distributions across clusters. Certain clusters may consistently dominate with high or low weights, particularly when they group overly simple or excessively difficult samples. This imbalance risks selecting subsets dominated by a few clusters, reducing diversity and potentially impairing the model's generalization performance.

To address this, we introduce a diversity loss $\mathcal{L}_D$, which minimizes the variance of average weights across clusters:

$$\mathcal{L}_D = \text{Var}(\{\bar{w}_1, \bar{w}_2, \ldots, \bar{w}_m\}), \quad \bar{w}_i = \frac{1}{n_i} \sum_{k=1}^{n_i} w_{ik}, \tag{2}$$

where $\text{Var}(\cdot)$ denotes variance, and $\bar{w}_i$ is the average weight of samples in the $i$-th cluster. Only clusters with samples in the current batch are included in the computation.

This loss encourages balanced weight distribution across clusters, promoting inter-cluster diversity while preserving intra-cluster variation. For clusters with high average weights (e.g., red an-d green

in Figure 2 top-left), $\bar{w}_i$ is reduced, while for clusters with low weights (e.g., purple and blue), $\bar{w}_i$ is increased. This prevents overrepresentation of any single cluster in the final selection, ensuring a more diverse subset.

Notably, $\mathcal{L}_D$ only acts on the cluster-level mean values $\{\bar{w}_i\}$, and does not penalize the variance within each cluster, thereby preserving the intra-cluster ranking of sample importance. By jointly optimizing importance and diversity losses, CoIDO scores reflect both the importance and diversity of samples, ensuring a balanced and representative data selection.

## 3.2 Coupled Optimization

Importance and diversity somehow inherently conflict in data selection. Prior methods like SELF-FILTER [20] and TIVE [17] address these objectives in separate optimization stages but often fail to achieve an optimal balance. Unlike these approaches, our method considers both importance and diversity in modeling $w_{ik}$. However, direct summation requires heuristic tuning of scalar weights between objectives, which can be sensitive to scale differences and lead to unstable or biased optimization. Moreover, fixed weights cannot adapt to varying task uncertainties or objective difficulties during training, potentially resulting in suboptimal trade-offs between importance and diversity. To address this, we propose a dynamic and automatic balancing mechanism during training.

We formulate this problem using a *maximum likelihood estimation* (MLE) framework under task-specific uncertainty. Each loss term is treated as the *negative log-likelihood* of a probabilistic model, where learnable parameters capture the inherent uncertainty or noise in optimizing each objective. This concept, termed *homoscedastic uncertainty* [22], assumes that uncertainty is constant for a given objective but varies across objectives, aligned well with our multimodal scenario.

We introduce $\sigma_I$ and $\sigma_D$ as learnable parameters representing the uncertainty for the importance and diversity objectives, respectively. Let $\theta$ denote the model parameters, $\mathbf{y}$ the outputs of the target MLLM, and $\mathbf{w}$ the CoIDO scores from the scorer. Using the multi-task likelihood framework, the total likelihood is modeled as:

$$\log p(\mathbf{y}, \mathbf{w} \mid \theta, \sigma_I, \sigma_D) = \sum\nolimits_{i,k} \log p(y_{ik} \mid x_{ik}, \theta, \sigma_I) + \sum\nolimits_i \log p(\bar{w}_i \mid \theta, \sigma_D). \tag{3}$$

**Importance Objective**. We begin with the importance objective $\mathcal{L}_I$, formulated within an instance-weighted likelihood framework. For a sample $(x_{ik}, y_{ik})$, we directly introduce a temperature parameter $\sigma_I$ and a sample-wise score $w_{ik}$ to scale logits, yielding a weighted Boltzmann (Gibbs) distribution [43]:

$$p(y_{ik} \mid x_{ik}, \theta, \sigma_I, w_{ik}) = \text{Softmax}\left(\frac{w_{ik}}{\sigma_I^2} f_\theta(x_{ik})\right), \tag{4}$$

where $\sigma_I$ controls the sharpness of the distribution, $f_\theta(x)$ is the model output, and $w_{ik}$ adjusts the sample's importance. Taking the negative log-likelihood yields:

$$-\log p = -\frac{w_{ik}}{\sigma_I^2} f_c + \log \sum\nolimits_j \exp\left(\frac{w_{ik}}{\sigma_I^2} f_j\right), \tag{5}$$

where $f_c$ is the ground-truth logit of class $c$. Defining $\alpha = \frac{w_{ik}}{\sigma_I^2}$, the second term in Equation 5 can be approximated as:

$$g(\alpha) = \log \sum\nolimits_j \exp(\alpha f_j) = \alpha \log S + \log \sum\nolimits_j p_j^\alpha, \tag{6}$$

where $S = \sum_j e^{f_j}$ and $p_j = e^{f_j}/S$. Applying a second-order Taylor expansion of $\log \sum_j p_j^\alpha$ around $\alpha = 1$:

$$\log \sum\nolimits_j p_j^\alpha \approx \log \sum\nolimits_j p_j - (\alpha - 1)H(p) + O((\alpha - 1)^2). \tag{7}$$

Given that $\sum_j p_j = 1$, the zero-order term is eliminated, and $O((\alpha - 1)^2)$ refers to second-order infinitesimals. Besides, $H(p) = -\sum_j p_j \log p_j$ is the entropy. Substituting this approximation back to $g(\alpha)$, we have:

$$g(\alpha) \approx \alpha \log S - (\alpha - 1)H(p). \tag{8}$$

We justify why the first-order error term $(\alpha - 1)H(p)$ can be safely neglected in practice: According to the definition of entropy, $H(p)$ reaches its maximum value $\log C$ when the output distribution is

uniform over $C$ classes. However, in practice, model outputs are highly concentrated: the effective number of candidate tokens $T$ is typically much smaller than the vocabulary size $C$, reducing the upper bound of $H(p)$ to $\log T$.

Empirical studies [44, 45] suggest $T$ typically ranges from 5 to 10 for LLMs, keeping $H(p)$ low. Combined with the fact that $(\alpha - 1)$ is close to zero, the contribution of the first-order term to the gradient is negligible. Hence, the final approximation of the negative log-likelihood for a single sample simplifies to:

$$-\log p(y_{ik} \mid x_{ik}, \theta, \sigma_I, w_{ik}) \approx \frac{w_{ik}}{\sigma_I^2} \cdot \text{CE}(y_{ik}, \hat{y}_{ik}). \tag{9}$$

For a minibatch of $b$ samples from $m$ clusters (each with $n_i$ samples), the final importance loss in optimizations becomes:

$$-\sum_{i,k} \log p(y_{ik} \mid x_{ik}, \theta, \sigma_I) = \frac{1}{\sigma_I^2}\mathcal{L}_I + \log \sigma_I. \tag{10}$$

The $\log \sigma_I$ term arises from the normalization constant of the Softmax-based likelihood under temperature scaling, as a direct consequence of MLE.

**Diversity Objective**. Following the multi-task learning framework from Kendall et al. [22], we interpret the diversity loss probabilistically. We consider a regression perspective by modeling the cluster-level mean weights $\bar{w}_i$ as Gaussian-distributed random variables with a common mean $\mu$ and variance $\sigma_D^2$:

$$p(\bar{w}_i|\theta, \sigma_D) = \mathcal{N}(\bar{w}_i; \mu, \sigma_D^2). \tag{11}$$

In this formulation, the variance $\sigma_D^2$ reflects the homoscedastic uncertainty of the diversity objective, acting as a learned task-dependent weight that downscales $\mathcal{L}_D$ when the diversity signal is noisier or less reliable. The negative log-likelihood for this regression objective thus takes the form:

$$-\log p(\bar{w}_i \mid \theta, \sigma_D) = \frac{(\bar{w}_i - \mu)^2}{2\sigma_D^2} + \frac{1}{2}\log(2\pi\sigma_D^2). \tag{12}$$

By considering all clusters and letting the $\mu$ be the mean of $\bar{w}_i$, summing across all clusters, and omitting constant terms, we arrive at the diversity loss expressed as:

$$-\sum_i \log p(\bar{w}_i \mid \theta, \sigma_D) = \frac{1}{2\sigma_D^2}\mathcal{L}_D + \log \sigma_D, \tag{13}$$

where $\mathcal{L}_D = \text{Var}(\bar{w}_1, \ldots, \bar{w}_c)$ is exactly our diversity loss derived in Section 3.1.

**Coupled Objective**. Combing Equation 10 and Equation 13, we obtain our final optimization goal:

$$\mathcal{L}_{\text{total}} = \frac{1}{\sigma_I^2}\mathcal{L}_I + \frac{1}{2\sigma_D^2}\mathcal{L}_D + \log \sigma_I + \log \sigma_D. \tag{14}$$

In Equation 14, $\sigma_I$ and $\sigma_D$ regulate the *homoscedastic uncertainty* associated with the objectives of importance and diversity, facilitating the balancing and adaptive optimization of the selection. At inference time, we discard the uncertainty parameters and directly use the trained CoIDO Scorer to assign selection scores to all candidate samples. To construct the final subset, for each downstream task, we select the top-$\gamma$ fraction of samples with the lowest (i.e., most important-and-diverse) CoIDO scores within that task's candidate pool. This ensures the selected set maintains balanced task coverage while preserving the optimization goals of importance and diversity.

## 4 Experiments

**Model & Implementation**. We conduct experiments on the `LLaVA-665K` visual instruction tuning dataset [13], using `LLaVA-1.5-7B-LoRA` [3] as the target MLLM. Our framework trains on $p = 20\%$ of the dataset, sampled randomly, with training configuration aligned with the standard `LLaVA`. The CoIDO Scorer is a four-layer MLP. More implementation details are in Appendix B.2.

**Evaluation Tasks**. Following the previous work [19, 21], we evaluate CoIDO with a wide range of multimodal benchmarks that test different capabilities of MLLMs. The benchmarks include: 1)

Table 1: Overall performance and efficiency comparison of selection approaches across various multimodal evaluation benchmarks, with the best measures in bold and the second-best underlined.

| Method | VQAv2 | GQA | VizWiz | SQA-I | TextVQA | POPE | MME | MMBench en | MMBench cn | LLaVA-Bench | *Rel.* (%) | MLLM Training Data Cost (%) | Total FLOPs |
|---|---|---|---|---|---|---|---|---|---|---|---|---|---|
| Full Data | 79.1 | 63.0 | 47.8 | 68.4 | 58.2 | 86.4 | 1476.9 | 66.1 | 58.9 | 67.9 | 100 | ＼ | 10.2E |
| **Model-free Methods** | | | | | | | | | | | | | |
| Random | 75.9 | 59.3 | 43.6 | 68.6 | 55.3 | 85.9 | 1461.0 | 60.3 | 53.3 | 64.5 | 95.1 | ＼ | ＼ |
| CLIP-Score [29] | 73.4 | 51.4 | 43.0 | 65.0 | 54.7 | 85.3 | 1331.6 | 55.2 | 52.0 | 66.2 | 91.2 | ＼ | ＼ |
| EL2N [32] | 76.2 | 58.7 | 43.7 | 65.5 | 53.0 | 84.3 | 1439.5 | 53.2 | 47.4 | 64.9 | 92.0 | ＼ | ＼ |
| Perplexity [33] | 75.8 | 57.0 | 47.8 | 65.1 | 52.8 | 82.6 | 1341.4 | 52.0 | 45.8 | 68.3 | 91.6 | ＼ | ＼ |
| SemDeDup [30] | 74.2 | 54.5 | 46.9 | 65.8 | 55.5 | 84.7 | 1376.9 | 52.2 | 48.5 | 70.0 | 92.6 | ＼ | ＼ |
| D2-Pruning [31] | 73.0 | 58.4 | 41.9 | 69.3 | 51.8 | 85.7 | 1391.2 | 65.7 | 57.6 | 63.9 | 94.8 | ＼ | ＼ |
| Self-Sup [30] | 74.9 | 59.5 | 46.0 | 67.8 | 49.3 | 83.5 | 1335.9 | 61.4 | 53.8 | 63.3 | 93.4 | ＼ | ＼ |
| **Model-involved Methods** | | | | | | | | | | | | | |
| Self-Filter [20] | 73.7 | 58.3 | **53.2** | 61.4 | 52.9 | 83.8 | 1306.2 | 48.8 | 45.3 | 64.9 | 90.9 | 100 | 31.2E |
| TIVE♣♦ [17] | 76.0 | 58.4 | 44.6 | 69.8 | 53.3 | 85.7 | 1448.4 | **66.9** | **58.7** | 63.4 | 96.7 | 100+8 | 11.7E |
| ICONS♣◇ [19] | 77.0 | **60.4** | 45.5 | **70.4** | 54.5 | **86.1** | 1447.7 | 64.6 | 54.0 | 66.9 | 97.1 | 100+5+2.2 | 12.6E |
| COINCIDE [21] | 76.5 | 59.8 | 46.8 | 69.2 | **55.6** | **86.1** | **1495.6** | 63.1 | 54.5 | 67.3 | 97.4 | 100 | 4.9E |
| CoIDO (Ours) | **77.2** | **60.4** | 47.1 | 69.4 | **55.6** | 85.4 | 1450.2 | 63.8 | 56.7 | **70.1** | **98.2** | **20** | **4.2E** |

♣ These methods need additional data to train the target MLLM for warm-up or as a reference set. ♦ reproduced by us, as the relevant LoRA results are not included in the paper. ◇ Reproduced with the optimal subset `LLaVA-ICONS-133k` released by ICONS within our unified evaluation framework (see an examination of the evaluation framework in Appendix B.3), corresponding to the original reports *Rel.* = 98.6%.

visual question answering: VQAv2 [46], GQA [47], VizWiz [48]; 2) knowledge-grounded QA: ScienceQA [49]; 3) Optical Character Recognition (OCR): TextVQA [50]; 4) hallucination: POPE [51]; 5) multiple-choice: MME [52], MMBench [53]; 6) free-form generation: LLaVA-Bench (In-the-Wild) [3]. More information about these benchmarks is provided in Appendix B.3. Since each evaluation benchmark has a different scale, we compute average relative performance, denoted as *Rel.*, across the ten benchmarks to assess the level of generalization. Relative performance on each benchmark is defined as: (subset model performance / full-data model performance) × 100%.

## 4.1 Overall Performance and Efficiency Comparison

**Metrics**. We introduce a key efficiency metric: *MLLM Training Data Cost*, which quantifies the proportion of data used to train the selection model relative to model fine-tuning. A 100% MLLM Training Data Cost implies no reduction in resource usage compared to full-data training. In addition, we report *total FLOPs*, derived as the product of parameter volume and training sample number.

**Baselines**. Table 1 compares the proposed CoIDO to two baseline categories: **1)** *model-free methods* include random sampling and CLIP-Score [29], which rely on simple criteria without resorting to the target MLLM. While computationally efficient (zero MLLM Training Data Cost and low FLOPs), their limited understanding of the data compromises effectiveness and transferability. **2)** *model-involved methods* [20, 21, 17, 19] require fine-tuning the target MLLM during data selection, resulting in significant computational costs (usually 100% MLLM Training Data Cost). Gradient-based methods like TIVE and ICONS also require additional warm-up data (8% and 5%, respectively), with ICONS further relying on benchmark validation sets (2.2% of the total data) for gradient computations. While this provides high-quality, task-specific data, it is often impractical in real-world scenarios. COINCIDE uses a smaller MLLM TinyLLaVA-2B as the reference model, but the entire data traversal still incurs a high training cost compared to ours. See more details of these baselines in Appendix B.1.

**Performance & Efficiency Results**. As shown in Table 1, CoIDO achieves a *Rel.* of 98.2% while using only $p = 20$ (%) of the data for framework training. Compared to model-free methods, CoIDO provides significantly higher performance and robustness. Against model-involved methods, it matches the SOTA performance of ICONS but reduces MLLM Training Data Cost to just 20%. Unlike ICONS, CoIDO does not require additional validation sets, simplifying the process while ensuring optimal performance. Furthermore, CoIDO achieves the lowest total computational cost, requiring only 4.2E (ExaFLOPs), outperforming all other methods in efficiency.

## 4.2 Ablation Study and Design Choices

We perform ablation studies to confirm the design choice of CoIDO, focusing on coupled optimization strategies and the scorer architectures. See more ablation studies in Appendix C.

**Optimization Methods**. CoIDO employs a coupled optimization strategy (Section 3.2), which jointly optimizes data importance and diversity via integrated learning objectives. To assess its effectiveness, we test several optimization strategies, summarized in Table 2. As a baseline, we first adopt a conventional approach that optimizes the importance loss $\mathcal{L}_I$ alone during training, omitting the diversity loss $\mathcal{L}_D$. Diversity is added only at the selection stage via a standalone procedure. This approach achieves the lowest *Rel.*=89%. We also experiment with alternative formulations, including simple summation of the two loss terms and weighted combinations using *learnable* coefficients (e.g., $\lambda$ and $1 - \lambda$). Among all configurations, our homoscedastic uncertainty optimization proves most effective (the best in most cases, while performing rather close to the best measure).

Table 2: Ablations of optimization methods (the best in bold and the second-best underlined).

| Loss Function | VQAv2 | GQA | Vizwiz | SQA-I | TextVQA | POPE | MME | MMBench(en) | MMBench(cn) | LLAVA-B | *Rel.* (%) |
|---|---|---|---|---|---|---|---|---|---|---|---|
| $\mathcal{L}_I$ | **77.9** | 48.9 | 44.6 | 59.7 | 52.5 | **86.2** | 1393.5 | 51.1 | 44.9 | 64.9 | 89.0 |
| $\mathcal{L}_I + \mathcal{L}_D$ | 74.5 | 55.8 | 46.4 | 67.3 | 52.6 | 83.5 | 1339.7 | 57.0 | 50.9 | 62.3 | 92.0 |
| $\lambda\mathcal{L}_I + (1-\lambda)\mathcal{L}_D$ | 76.1 | 59.4 | 46.8 | 68.7 | 54.4 | 85.2 | **1465.6** | 60.5 | 54.0 | 64.6 | 95.9 |
| Ours | 77.2 | **60.4** | **47.1** | **69.4** | **55.6** | 85.4 | 1450.2 | **63.8** | **56.7** | **70.1** | **98.2** |

**Scorer Architecture**. To explore alternatives to the MLP-based CoIDO Scorer, we evaluate two additional designs: (1) a *Transformer-based Scorer* with two standard Transformer blocks, and (2) an *attention-only Scorer* using self-attention and cross-attention mechanisms applied directly to extracted features. The Transformer-based variant combines stacked attention and feedforward layers for enhanced capacity, while the attention-only design explicitly captures intra- and inter-sample relationships. Figure 3 compares their performance (*Rel.*) and training-time FLOPs. The strong performance of the MLP-based scorer can be attributed to the expressiveness of the extracted multimodal features (e.g., CLIP Score and ImageReward), which already encode rich semantic alignment and quality cues, thereby reducing the need for additional modeling complexity.

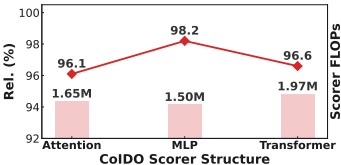

Figure 3: Ablations of different CoIDO Scorer architectures.

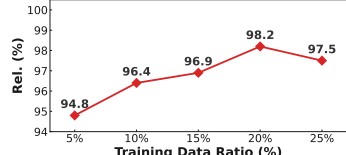

Figure 4: Comparison of different training data ratios $p\%$.

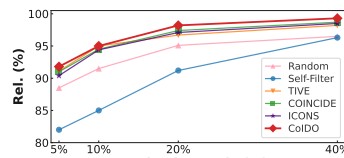

Figure 5: Performance vs. selection ratios $\gamma$.

### 4.3 Parameter Sensitivity Study

We analyze 1) the *training data ratio* ($p\%$) and 2) the *selection ratio* ($\gamma$ defined in the task description) that denotes the proportion of data selected for training. See more parameter analyses in Appendix C.

**Training Data Ratio**. CoIDO enables effective data selection using a small fraction of the training data. To identify an optimal $p\%$, we gradually increase $p\%$ starting from 5%. As shown in Figure 4, performance is modest when $p < 10$ (%), but improves clearly beyond this threshold. Notably, *Rel.* stabilizes after $p > 20$ (%), indicating that 20% data is sufficient to capture the dataset distribution. Based on these findings, we set $p = 20$ (%) as the default for all experiments.

**Selection Ratio**. We evaluate CoIDO and baselines across selection ratios ($\gamma$) ranging from 5% to 40%. Figure 5 shows that CoIDO consistently outperforms the competitors across this range. However, as $\gamma$ approaches 40%, the performance gap narrows since most high-quality samples in LLaVA-665K would be identified by all methods. When the selection ratio exceeds 50%, the impact of randomness increases, and even a random selection strategy may yield strong results due to the inclusion of a large portion of the dataset (this shows a necessity of reducing visual instruction tuning data in practice). Hence, we focus on selection ratios between 5% and 40%, where differences in selection methods are most distinct and provide meaningful insights into method effectiveness. Notably, CoIDO consistently achieves the highest *Rel.* across all selection ratios.

## 4.4 Generalizability and Transferability

In order to further explore the potential of CoIDO, we conduct experiments about **generalizability** and **transferability**. We define **generalizability** as the ability of the proposed data selection framework to be directly applied to other models or datasets. **Transferability**, on the other hand, measures whether a CoIDO scorer trained on one domain can be reused to select informative data in another, out-of-domain corpus. To evaluate these two aspects, we conduct experiments on the `Vision-Flan` dataset, a large-scale human-annotated visual instruction-tuning benchmark containing over 200 diverse vision–language tasks. Additional results on other models and datasets (e.g., `LLaVA-13B` and `LLaVA-150K`) are presented in Appendix C.4 and C.5. Both results on generalizability and transferability are summarized in Table 3.

Table 3: Performance of CoIDO on the `Vision-Flan` dataset (20% data selection). $^\ddagger$ CoIDO scorer trained on `LLaVA-665K` and applied to `Vision-Flan` (out-of-domain transfer).

| Model / Setting | VQAv2 | GQA | VizWiz | SQA | POPE | TextVQA | MME | MMBench(en) | MMBench(cn) | LLaVA-B | *Rel.* (%) |
|---|---|---|---|---|---|---|---|---|---|---|---|
| Full Fine-tune | 74.5 | 47.1 | 52.8 | 61.8 | 46.4 | 85.7 | 1480.6 | 40.2 | 46.2 | 38.2 | 100.0 |
| Random | 74.6 | 44.3 | 50.0 | 59.8 | 40.9 | 81.3 | 1407.1 | 49.2 | **48.3** | 33.6 | 97.8 |
| CoIDO | **75.7** | 45.1 | **53.5** | 62.3 | **45.3** | 82.8 | 1452.9 | **52.0** | 46.8 | 37.6 | 102.1 |
| CoIDO$^\ddagger$ | **75.7** | **46.8** | 53.3 | 66.2 | 42.1 | **85.5** | **1486.1** | 51.4 | 47.3 | **40.8** | **103.7** |

**Generalizability**. As shown in the first three rows of Table 3, CoIDO consistently surpasses random selection and even slightly outperforms full fine-tuning under the same 20% budget, achieving a relative score of 102.1%. The gains are especially notable on reasoning-intensive and perception-heavy benchmarks such as VizWiz and POPE, demonstrating that CoIDO is capable of identifying semantically rich and instruction-relevant samples even within large human-curated datasets. These results confirm CoIDO's strong generalizability which can be applied across different datasets.

**Transferability**. To further assess knowledge transfer, we reuse the CoIDO scorer trained on `LLaVA-665K` directly for data selection on `Vision-Flan`, without retraining or domain adaptation. As indicated by the last row of Table 3, the scorer trained on `LLaVA-665K` achieved even better results on `Vision-Flan` than the scorer trained directly on `Vision-Flan` itself. We believe this is because `LLaVA-665K` is a significantly larger and more diverse dataset, allowing the scorer to learn a more generalizable notion of sample difficulty and importance. This enables it to transfer effectively across domains, even outperforming in-distribution scorers trained on smaller datasets. This indicates that CoIDO can seamlessly be transferred to new domain datasets with similar distributions, demonstrating its transferability.

## 5 Conclusion and Future work

In this paper, we present an efficient data selection framework for visual instruction tuning in multimodal large language models (MLLMs). By coupling the optimization of data importance and diversity, and employing a lightweight scorer to learn data distribution from a small subset, our CoIDO framework achieves state-of-the-art performance while significantly reducing the amount of required training data and computational cost. This scalable and practical approach provides a valuable contribution to improving efficiency in MLLM fine-tuning.

Future work includes exploring co-training with smaller, proxy MLLMs, such as `TinyLLaVA`, to further reduce training costs and validating scalability on larger MLLMs like `LLaVA-13B`. Additional directions involve dynamic data selection for evolving datasets and training progress and extending the framework to other multimodal scenarios, e.g., audio LLMs and video LLMs.

## 6 Acknowledgement

This work was supported by the Zhejiang Provincial Natural Science Foundation (No. LD24F020015), Pioneer R&D Program of Zhejiang (No. 2024C01021), and NSFC (Nos. U24A20254, 62471168, 62572075, ). Jinpeng Chen was additionally supported by BNSF (No. L233034) and Fundamental Research Funds for BUPT (No. 2025TSQY01).

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

# A More Technical Details

## A.1 Details in Feature Extraction and Spectral Clustering

**Feature Extraction**. Before our training process, we derive four categories of basic features for each training data sample. These features are intended to encapsulate the quality of images, text, and the alignment of image-text. A comprehensive overview of these basic features is provided in Table 4, while the exact prompt employed in our LLM-based evaluator is depicted in Figure 6.

Table 4: Explanations of our basic features extracted before the training process.

| Indicator | Explanation |
|---|---|
| *Image-Reward Score* [38] | This score from the ImageReward model [38], which assesses the quality of images generated from text prompts by aligning them with human preferences, focusing on aspects such as text-image alignment, visual fidelity, and overall aesthetic appeal. |
| *LLM Score* | The *LLM Score* represents a robust measure employed by a language model to assess the response's quality, specifically in terms of grammar, spelling, and fluency. This score indicates how well the generated caption aligns with the model's language capabilities. In our approach, we utilize DeepSeek-V3 [54] for text evaluation, with the prompts illustrated in Figure 6. |
| *CLIP Features* | Vision-language features in low-dimensional space obtained by encoding images with ViT from CLIP [39] and text with Llama2 [55], followed by conducting unsupervised dimensionality reduction. |
| *CLIP Score* [29] | This score is the Cosine similarity between the embedding of an image and that of its corresponding text. This metric serves as an indicator of the semantic alignment between visual and textual modalities. It quantifies how well the caption represents the visual content and is used to assess the coherence between image-text pairs. |

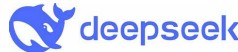

**System Prompt:**

We would like to request your feedback on the performance of AI assistant in response to the user's questions in the conversation displayed following.

Conversation: [text]

**User Prompt:**

Please rate according to the content of the responses to the questions. The assistant should receive a score on a scale of 0 to 10, where a higher score indicates higher level of the content. Please first output a single line containing the value indicating the scores. In the subsequent line, please provide a comprehensive explanation of your evaluation, avoiding any potential bias.

Figure 6: The prompt is analogous to that used in ALPAGASUS [15]. We employ DeepSeek-V3 [54] to evaluate the quality of the text.

**Spectral Clustering**. We employ spectral clustering to group training data samples based on their basic features. The process begins by constructing a similarity matrix from the input basic features, which are illustrated in Table 7. We utilize Cosine similarity to quantify the pairwise relationships between data samples. To focus on the most relevant connections and manage computational complexity, a $K$-Nearest Neighbors ($K$NN) graph is then built upon this similarity matrix, effectively

creating a sparse representation of the data's affinity structure. The spectral clustering algorithm is subsequently applied to this graph, partitioning the data into a predefined number of clusters by analyzing the spectrum (eigenvalues) of the graph Laplacian. This method allows for the identification of clusters based on connectivity rather than solely relying on compactness.

Compared to $K$-means, spectral clustering offers distinct advantages for analyzing complex image-text datasets. Firstly, spectral clustering is capable of identifying clusters with non-convex shapes, which $K$-means often struggles with due to its assumption of spherical cluster geometry. The rich semantic relationships in image-text data often manifest as intricate structures in the feature space that are better captured by spectral methods. Secondly, by operating on a graph representation, spectral clustering can reveal underlying manifold structures within the data. This is particularly beneficial for image-text embeddings, where meaningful relationships might not be best represented by simple Euclidean distances to cluster centroids, but rather by the connectivity patterns between samples. This allows for a more nuanced understanding of how different image-text pairs relate to one another, leading to potentially more coherent and semantically meaningful groupings.

### A.2 Pseudocode of the Framework Training

We provide detailed pseudocode for our CoIDO framework in Algorithm 1.

---

**Algorithm 1** CoIDO: Training the CoIDO Scorer

---

**Require:** Instruction dataset $\mathcal{D}$, MLLM $f$, Scorer, sampling ratio $p\%$, clusters $M$, batch size $s$
 1: Randomly sample $D_r \subset \mathcal{D}$, $|D_r| \leftarrow p\%|\mathcal{D}|$
 2: Extract basic features $\mathbf{x}_j$ for each $x_j$ in $D_r$
 3: Cluster $D_r$ into $M$ groups using spectral clustering
 4: **for** each training step **do**
 5:    Sample minibatch $B \leftarrow \{x_j\}_{j=1}^s \sim D_r$
 6:    **for** each $x_j$ in $B$ **do**
 7:       Compute prediction $\hat{y}_j \leftarrow f(x_j)$
 8:       Compute loss $\mathcal{L}_j \leftarrow \mathrm{CE}(y_j, \hat{y}_j)$
 9:       Compute weight $w_j \leftarrow \mathrm{Scorer}(\mathbf{x}_j)$
10:    **end for**
11:    Compute $\mathcal{L}_I \leftarrow \sum_{j=1}^s w_j \mathcal{L}_j$
12:    **for** cluster $C_i$ in this minibatch , $i = 1, 2, \ldots, m$ **do**
13:       Compute average weight $\bar{w}_i \leftarrow \mathrm{Mean}(w_j, \; x_j \in C_i)$
14:    **end for**
15:    Compute diversity loss $\mathcal{L}_D \leftarrow \mathrm{Var}(\bar{w}_1, \ldots, \bar{w}_m)$
16:    Compute total loss $\mathcal{L}_{\text{total}} \leftarrow \frac{1}{\sigma_I^2}\mathcal{L}_I + \frac{1}{2\sigma_D^2}\mathcal{L}_D + \log \sigma_I + \log \sigma_D$
17:    Update $f$, Scorer, and $\sigma_I, \sigma_D$ using $\mathcal{L}_{\text{total}}$
18: **end for**

---

## B   More Experimental Setting Details

### B.1   Baseline Description

**Model-free Methods**. Model-free methods select training data without relying on the training or inference of large multimodal language models (MLLMs). Instead, they utilize static heuristics or precomputed metrics to evaluate sample quality or diversity. Due to the absence of target MLLM, the results of such methods are usually not satisfactory. These baselines including

- RANDOM SAMPLING: This baseline method involves randomly selecting samples from the dataset without any specific criteria. It serves as a reference point to assess the effectiveness of more sophisticated selection strategies.

- CLIP-SCORE [29]: Utilizes the Cosine similarity between image and text embeddings generated by the CLIP model to measure the semantic alignment of image-text pairs. Higher scores indicate better alignment.

- EL2N [32]: Calculates the L2 norm of the loss for each sample during the initial epochs of training. Samples with higher EL2N scores are considered more informative, as they are harder to learn early on and are thus prioritized for training.

- PERPLEXITY [33]: Employs a language model to compute the perplexity of textual responses. Lower perplexity scores suggest that the text is more predictable and, by extension, of higher quality.

- SEMDEDUP [30]: Implements semantic deduplication by identifying and removing semantically similar or duplicate samples from the dataset, thereby enhancing diversity and reducing redundancy.

**Model-involved Methods**. Model-involved methods incorporate the training of MLLMs into the data selection process. These approaches often involve fine-tuning the MLLM, estimating gradient-based influence, or leveraging internal representations of MLLMs to evaluate the utility of each sample. Although these methods are typically more computationally expensive, they tend to outperform model-free methods by capturing more informative signals about task relevance, data quality, and alignment with downstream objectives. These baselines including

- SELF-FILTER [20]: Trains a scoring network alongside the MLLM to evaluate the quality of instruction samples. By filtering out low-quality data based on model feedback, it aims to enhance the overall training efficiency and performance.

- TIVE [17]: Estimates the value of each task-instance pair by analyzing gradient-based influence scores. This method selects samples that are expected to have the most significant impact on model performance, thereby reducing redundancy in the training data.

- ICONS [19] (Influence Consensus): Aggregates influence scores across multiple tasks to identify samples that consistently contribute positively to performance. By focusing on these consensus samples, ICONS facilitates efficient multi-task learning with a compact dataset.

- COINCIDE [21]: Combines multimodal features and task relevance to select representative samples that capture the essential characteristics of vision-language data. This approach aims to improve the model's generalization and multi-task learning capabilities.

## B.2 Training Configuration of Our Method

We implement our method based on the LLaVA-1.5-7B architecture with LoRA tuning [56]. All experiments are conducted using eight A5000 GPUs with 24GB of memory. The fine-tuning process utilizes LoRA [56] across 2 epochs, with each GPU processing a batch size of 8 and a learning rate set at 2e-5. The Adam optimizer is applied, featuring no weight decay, alongside a Cosine learning rate schedule and a warmup ratio of 3%.

Following LLaVA [3], we use CLIP-ViT-L-336px for visual embedding and Vicuna-7B for text encoding. Spectral clustering is performed on normalized features with the number of clusters set to $M = 20$. Two learnable uncertainty parameters, $\sigma_I$ and $\sigma_D$, are initialized to 0. The hidden layer size of our MLP is specified as 1536 for handling *CLIP features* and 3 for combining *LLM Score*, *CLIP Score*, and *Image-Reward Score*. Consistent with the original paper, we applied the same fine-tuning settings to the LLaVA-1.5-7B model for both LoRA and full fine-tuning.

## B.3 Evaluation Setup

**Benchmarks**. Our MLLM undergoes evaluation through ten diverse benchmarks designed to assess a range of capabilities, such as visual question answering, reasoning, robustness, and following instructions. A detailed overview of these benchmarks is provided in Table 5.

**Evaluation Framework**. In all experiments, we evaluate all benchmarks in strict accordance with the official guidelines and procedures of LLaVA-1.5-7B [13]. Table 6 presents the relative correctness of our unified evaluation framework. LLaVA-Bench was initially evaluated using gpt-3.5-turbo-0613. However, this model has since been deprecated by OpenAI. Based on discussions within the community[2] and comprehensive consideration, we switched to using gpt-4o-mini for evaluation.

---

[2] The related discussion on `https://github.com/EvolvingLMMs-Lab/lmms-eval/issues/294`.

Table 5: Descriptions of benchmarks used to evaluate the MLLM performance.

| Benchmark | Explanation |
|---|---|
| VQAv2 [46] | A widely-used benchmark for visual question answering, consisting of natural images paired with open-ended questions that require understanding visual content to produce correct answers. |
| GQA [47] | This benchmark evaluates models on questions that require reasoning about real-world images, emphasizing logical inference and relational reasoning capabilities. |
| VizWiz [48] | A visual question answering benchmark specifically designed for visually impaired users, containing challenging real-world images with questions often needing practical reasoning about ambiguous or incomplete visual content. |
| SQA-I [49] | Situated Question Answering on Images, a benchmark designed to assess how well models can handle questions requiring situational and contextual visual understanding. |
| TextVQA [50] | A benchmark for evaluating model capability in answering questions requiring reading and reasoning over textual information presented in images, such as signs or printed text. |
| POPE [51] | A visual understanding benchmark focused on identifying the presence or absence of specific objects or entities in images, emphasizing accuracy and robustness in object recognition. |
| MME [52] | Multimodal Evaluation, an aggregate benchmark that assesses comprehensive multimodal understanding by combining various tasks including reasoning, captioning, and classification across multiple modalities. |
| MMBench (en/cn) [53] | A multimodal benchmark assessing the performance of models in answering complex multimodal questions in both English (en) and Chinese (cn), highlighting multilingual and multimodal understanding capabilities. |
| LLaVA-Bench [3] | A tailored benchmark for evaluating multimodal large language models (MLLMs), focusing on complex reasoning and understanding capabilities from multimodal data, particularly image-text pairs. |

This change in evaluation framework is the primary reason for the notable difference observed in the metrics presented in Table 6.

To ensure the validity of our evaluation pipeline, we first verified that our unified evaluation framework accurately reproduces the official results of LLaVA-1.5-7B on all benchmarks. As shown in Table 6, the reproduced scores are consistently aligned with the original reports, despite the shift from gpt-3.5-turbo-0613 to gpt-4o-mini as the evaluation backend.

However, when reproducing the ICONS baseline, we observed a notable discrepancy: our evaluation yields a relative performance of 97.1%, while the original ICONS paper reports 98.6%. We emphasize that this difference does not stem from issues in our framework. Our result is obtained using the officially released subset (LLaVA-ICONS-133k) provided by ICONS and evaluated using the same scoring protocol applied to all methods. Given the consistency of our framework across all other benchmarks and baselines, we attribute this minor drop to changes in evaluation prompts, model versions, or post-processing details on ICONS's side that were not fully disclosed. Therefore, this discrepancy should not be interpreted as a flaw in our implementation, but rather as a natural outcome of aligning all methods within a consistent and transparent evaluation setup.

Table 6: Validation of the evaluation procedure.

| Evaluation Setting | VQAv2 | GQA | Vizwiz | SQA-I | TextVQA | POPE | MME | MMBench(en) | MMBench(cn) | LLaVA-B | *Rel.* (%) |
|---|---|---|---|---|---|---|---|---|---|---|---|
| Original Report | 79.1 | 63.0 | 47.8 | 68.4 | 58.2 | 86.4 | 1476.9 | 66.1 | 58.9 | 67.9 | 100.0 |
| Our Reproduction | 79.1 | 62.9 | 48.4 | 68.6 | 58.2 | 86.4 | 1478.1 | 66.6 | 58.9 | 69.4 | 100.4 |
| *Difference* ($\Delta$) | 0.0 | -0.1 | +0.6 | +0.2 | 0.0 | 0.0 | +1.2 | +0.5 | 0.0 | +1.5 | +0.4 |

Table 7: Feature components used in CoIDO Scorer training.

| Feature Composition | VQAv2 | GQA | Vizwiz | SQA-I | TextVQA | POPE | MME | MMBench(en) | MMBench(cn) | LLAVA-B | *Rel.* (%) |
|---|---|---|---|---|---|---|---|---|---|---|---|
| Only CLIP Features | 76.5 | 59.0 | 51.1 | 68.6 | 55.1 | 82.9 | 1423.8 | 60.1 | 53.1 | 66.1 | 96.3 |
| Only Scores | 77.0 | 59.6 | 47.4 | 66.4 | 54.7 | 81.4 | 1,456.9 | 61.9 | 55.5 | 68.4 | 96.4 |
| All (Ours) | 77.2 | 60.4 | 47.1 | 69.4 | 55.6 | 85.4 | 1450.2 | 63.8 | 56.7 | 70.1 | 98.2 |

Table 8: The number of clusters $M$ ($M$ is set to 20 by default in our main experiments).

| #Clusters | VQAv2 | GQA | Vizwiz | SQA-I | TextVQA | POPE | MME | MMBench(en) | MMBench(cn) | LLAVA-B | *Rel.* (%) |
|---|---|---|---|---|---|---|---|---|---|---|---|
| 5 | 74.3 | 56.7 | 47.1 | 64.7 | 52.1 | 85.9 | 1398.4 | 53.4 | 48.8 | 66.2 | 92.2 |
| 10 | 74.0 | 58.3 | 49.8 | 68.1 | 51.7 | 87.6 | 1415.8 | 59.3 | 49.7 | 62.6 | 94.2 |
| 20 | 77.2 | 60.4 | 47.1 | 69.4 | 55.6 | 85.4 | 1450.2 | 63.8 | 56.7 | 70.1 | 98.2 |
| 40 | 76.1 | 59.2 | 51.2 | 68.1 | 54.1 | 85.6 | 1512.4 | 60.7 | 54.9 | 66.3 | 97.4 |

Table 9: Application on full fine-tuning.

| Method | VQAv2 | GQA | Vizwiz | SQA-I | TextVQA | POPE | MME | MMBench(en) | MMBench(cn) | LLAVA-B | *Rel.* (%) |
|---|---|---|---|---|---|---|---|---|---|---|---|
| Full Data | 78.5 | 62.0 | 50.0 | 66.8 | 58.2 | 85.9 | 1510.7 | 64.3 | 58.3 | 65.4 | 100.0 |
| RANDOM | 74.5 | 56.5 | 48.5 | 66.5 | 54.9 | 83.1 | 1364.1 | 58.6 | 52.1 | 67.3 | 94.7 |
| Ours | 75.9 | 57.8 | 47.4 | 67.8 | 54.6 | 81.1 | 1402.9 | 60.6 | 55.0 | 68.8 | 96.1 |

# C More Experimental Results

## C.1 Feature Components

To ascertain the features employed in the training of CoIDO, an ablation study is performed focusing on the components of features utilized during this process. For this analysis, we employ only the *CLIP Features* or the three scores (*LLM Score*, *CLIP Score*, and *Image-Reward Score*). The findings are presented in Table 7. These results indicate that the omission of any feature results in a decline in overall performance.

## C.2 Number of Clusters

The cluster number $M$ is a crucial parameter in our framework, as the computation of diversity loss $\mathcal{L}_D$ for each batch depends on the number of clusters included in that batch ($m, m \leq M$). When $M$ substantially exceeds the batch size, it is highly probable that the samples in each batch originate from a limited subset of the clusters. Consequently, each time the diversity loss is evaluated, certain clusters may be excluded. Conversely, if $M$ is significantly smaller than the batch size, it is highly probable that samples from all clusters can be represented in each batch. However, choosing a too small for $M$ may, in turn, compromise the optimization of diversity. To choose the appropriate $M$, we experiment on different $M$ settings from 5 to 40, the results are reported in Table 8.

## C.3 Application on Full Fine-tuning

Table 1 presents a comparative analysis of LLaVA-1.5-7B-LoRA's performance. To assess the efficacy of supervised fine-tuning across the entire LLaVA-1.5-7B, we evaluated our selected subset, as detailed in Table 9. The findings indicate that, even with the direct fine-tuning of LLaVA, CoIDO remains effective, thereby confirming its generalizability.

## C.4 Application on Larger Model

Table 10 reports the evaluation results on the larger-scale LLaVA-13B-LoRA model. To further examine the scalability and generalizability of CoIDO, we apply the same data selection pipeline used for the 7B model to the 13B counterpart without any modification. The results demonstrate

that COIDO consistently outperforms existing baselines and maintains comparable performance to full-data fine-tuning under the same 20% budget. This confirms that COIDO generalizes effectively to larger model capacities, validating its robustness and adaptability across different parameter scales.

Table 10: Evaluation results on the LLaVA-13B-LoRA model under a 20% data selection ratio.

| Method | VQAv2 | GQA | VizWiz | SQA-I | TextVQA | POPE | MME | MMBench(en) | MMBench(cn) | LLaVA-B | *Rel.* (%) |
|---|---|---|---|---|---|---|---|---|---|---|---|
| Full Finetune | 80.0 | 63.3 | 58.9 | 71.2 | 60.2 | 86.7 | 1541.7 | 68.5 | 61.5 | 69.5 | 100.0 |
| Random | 76.7 | 60.5 | 48.0 | 68.8 | 57.7 | 84.8 | 1484.9 | 62.8 | 55.2 | 68.6 | 94.0 |
| ICONS | **77.9** | 60.5 | 47.7 | 74.0 | 57.3 | **87.4** | 1503.9 | **65.5** | **59.2** | 65.3 | 95.7 |
| COINCIDE | 77.8 | 60.4 | **51.6** | 70.0 | 58.6 | 87.1 | 1516.8 | 64.0 | 57.7 | 67.4 | 95.9 |
| **CoIDO (Ours)** | 77.5 | **61.5** | 51.1 | **74.2** | **58.9** | 85.8 | **1586.4** | 64.7 | 57.5 | **69.7** | **97.2** |

## C.5 Application on Smaller Dataset

Table 11 presents the evaluation results on the smaller-scale LLaVA-150K dataset. To further assess the generalizability of COIDO under limited data regimes, we directly apply our selection strategy without re-tuning any hyperparameters. As shown in Table 11, COIDO consistently achieves superior or comparable performance to full-data fine-tuning while using only 20% of the data, demonstrating its robustness and adaptability in low-resource scenarios.

Notably, the accuracies of GQA and TextVQA are consistently below 1% across all methods (close to random guessing), and thus omitted from the table. This behavior is expected because LLaVA-150K contains very few or no samples related to compositional reasoning (GQA) or OCR-centric tasks (TextVQA), rendering these benchmarks unreliable in this setting. Interestingly, for MMBench and MMBench(cn), both random selection and CoIDO obtain higher scores than full-data fine-tuning, which we attribute to the presence of noisy or low-quality samples in the complete dataset—selective fine-tuning helps mitigate overfitting and improves generalization to evaluation benchmarks.

Table 11: Performance of CoIDO on the LLaVA-150K dataset.

| Model | VQAv2 | VizWiz | SQA | POPE | MME | MMBench(en) | MMBench(cn) | LLaVA-B | Rel.(%) |
|---|---|---|---|---|---|---|---|---|---|
| Full Fine-tune | 55.2 | 45.5 | 57.6 | 57.9 | 1234.5 | 22.4 | 27.1 | 65.9 | 100.0 |
| Random | 50.0 | 44.8 | 53.4 | 54.2 | 1184.7 | 30.7 | 30.1 | 62.4 | 101.7 |
| **CoIDO (Ours)** | 49.5 | 47.1 | 58.2 | 56.1 | 1214.8 | 29.8 | 32.3 | 63.8 | 104.8 |

## C.6 Detailed Results for Main Paper Experiments

This section presents the complete per-benchmark evaluation results, as detailed in Tables 12, 13, and 14, corresponding to Figures 3, 4, and 5 in the main paper, respectively.

## C.7 Visual Analyses

To visually analyze the effectiveness of the scores outputed by the COIDO Scorer, we conduct a case study with results presented in Figure 7 and Figure 8.

# D Limitations

While COIDO demonstrates strong performance in data-efficient instruction tuning, several limitations remain. First, our approach relies on four handcrafted data features to represent each sample. Although the computation for these features is lightweight, this dependency may limit extensibility or introduce engineering overhead. Future work will explore whether an effective scorer can be learned from fewer or even latent representations to further streamline the pipeline.

Second, our experiments are currently limited to LLaVA-1.5-7B. The effectiveness of COIDO on larger models, such as LLaVA-13B or other recent MLLMs, is yet to be verified. Extending the evaluation to larger-scale models is crucial for assessing the scalability and generalizability of our approach. This will require significantly greater computational resources, but it is foreseeable that this task can be accomplished in a near future.

Third, although COIDO achieves high selection quality with significantly reduced computational cost, COINCIDE [21] also offers low FLOPs by leveraging a smaller reference model (TinyLLaVA-2B).

Table 12: Detailed results for different CoIDO Scorer architectures (MLP by default in our design).

| CoIDO Structure | VQAv2 | GQA | Vizwiz | SQA-I | TextVQA | POPE | MME | MMBench(en) | MMBench(cn) | LLAVA-B | Rel. (%) |
|---|---|---|---|---|---|---|---|---|---|---|---|
| Attention | 76.2 | 59.8 | 47.8 | 66.7 | 54.4 | 85.8 | 1468.0 | 60.5 | 53.9 | 65.8 | 96.1 |
| Transformer | 77.0 | 59.7 | 48.7 | 68.1 | 53.9 | 83.5 | 1439.7 | 63.4 | 55.9 | 64.2 | 96.6 |
| MLP | 77.2 | 60.4 | 47.1 | 69.4 | 55.6 | 85.4 | 1450.2 | 63.8 | 56.7 | 70.1 | 98.2 |

Table 13: Detailed results for different training data ratios (20% by default in our design).

| Ratio | VQAv2 | GQA | Vizwiz | SQA-I | TextVQA | POPE | MME | MMBench(en) | MMBench(cn) | LLAVA-B | Rel. (%) |
|---|---|---|---|---|---|---|---|---|---|---|---|
| 5% | 76.1 | 58.4 | 41.8 | 67.4 | 54.6 | 86.1 | 1449.1 | 61.9 | 53.9 | 65.4 | 94.8 |
| 10% | 77.0 | 59.4 | 46.4 | 66.8 | 54.2 | 83.3 | 1499.4 | 61.9 | 55.7 | 66.9 | 96.4 |
| 15% | 76.9 | 59.8 | 51.6 | 67.0 | 54.7 | 85.6 | 1438.6 | 61.1 | 55.0 | 64.1 | 96.9 |
| 20% | 77.2 | 60.4 | 47.1 | 69.4 | 55.6 | 85.4 | 1450.2 | 63.8 | 56.7 | 70.1 | 98.2 |
| 25% | 77.1 | 60.2 | 48.2 | 67.9 | 55.2 | 84.7 | 1438.9 | 63.4 | 56.5 | 68.0 | 97.5 |

Table 14: Detailed results for different selection ratios (20% by default in our design).

| Ratio | VQAv2 | GQA | Vizwiz | SQA-I | TextVQA | POPE | MME | MMBench(en) | MMBench(cn) | LLAVA-B | Rel. (%) |
|---|---|---|---|---|---|---|---|---|---|---|---|
| 5% | 73.5 | 55.1 | 42.5 | 68.1 | 51.2 | 84.2 | 1371.7 | 58.8 | 54.1 | 60.8 | 91.8 |
| 10% | 75.2 | 58.0 | 42.2 | 68.8 | 54.1 | 83.8 | 1440.3 | 60.0 | 54.6 | 68.8 | 94.8 |
| 20% | 77.2 | 60.4 | 47.1 | 69.4 | 55.6 | 85.4 | 1450.2 | 63.8 | 56.7 | 70.1 | 98.2 |
| 40% | 78.2 | 61.6 | 47.7 | 69.1 | 56.6 | 85.5 | 1484.1 | 62.5 | 56.1 | 67.3 | 98.3 |

This highlights a promising direction for future research — replacing the target MLLM with a small proxy model during scoring to further reduce training cost without compromising selection performance. This capability can be readily incorporated into CoIDO with additional development effort.

# E  Broader Impact

From a broader impact perspective, CoIDO promotes efficient fine-tuning by selecting compact, diverse, and informative subsets, which may reduce **energy consumption** and democratize access to instruction tuning for **resource-constrained users**. However, data selection algorithms inherently influence model behavior. Care must be taken to ensure fairness, representation, and robustness, especially when deploying models downstream in sensitive or high-stakes applications.

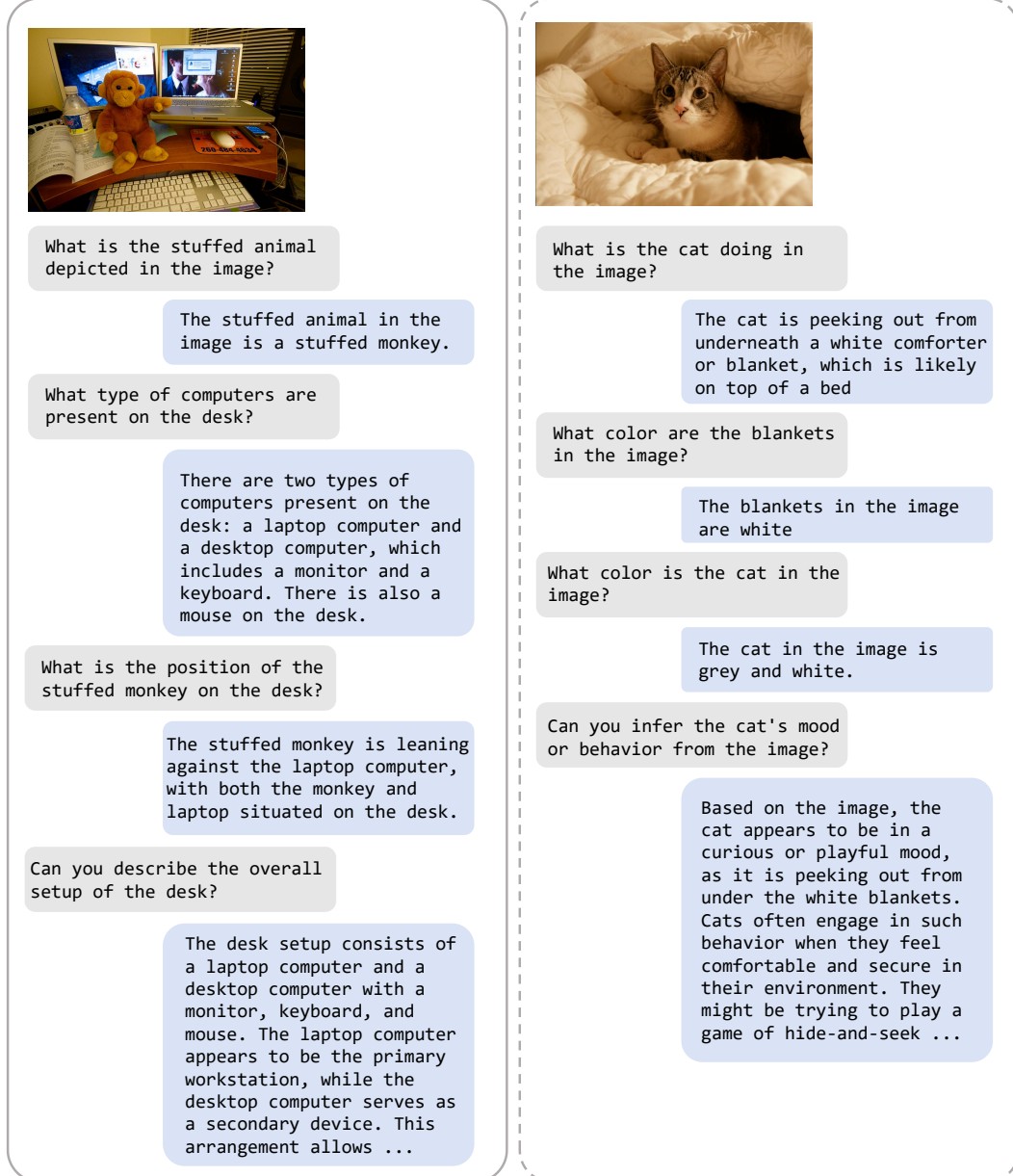

Figure 7: The best (left) and the worst (right) CoIDO scores samples from LLaVA-133K datasets.

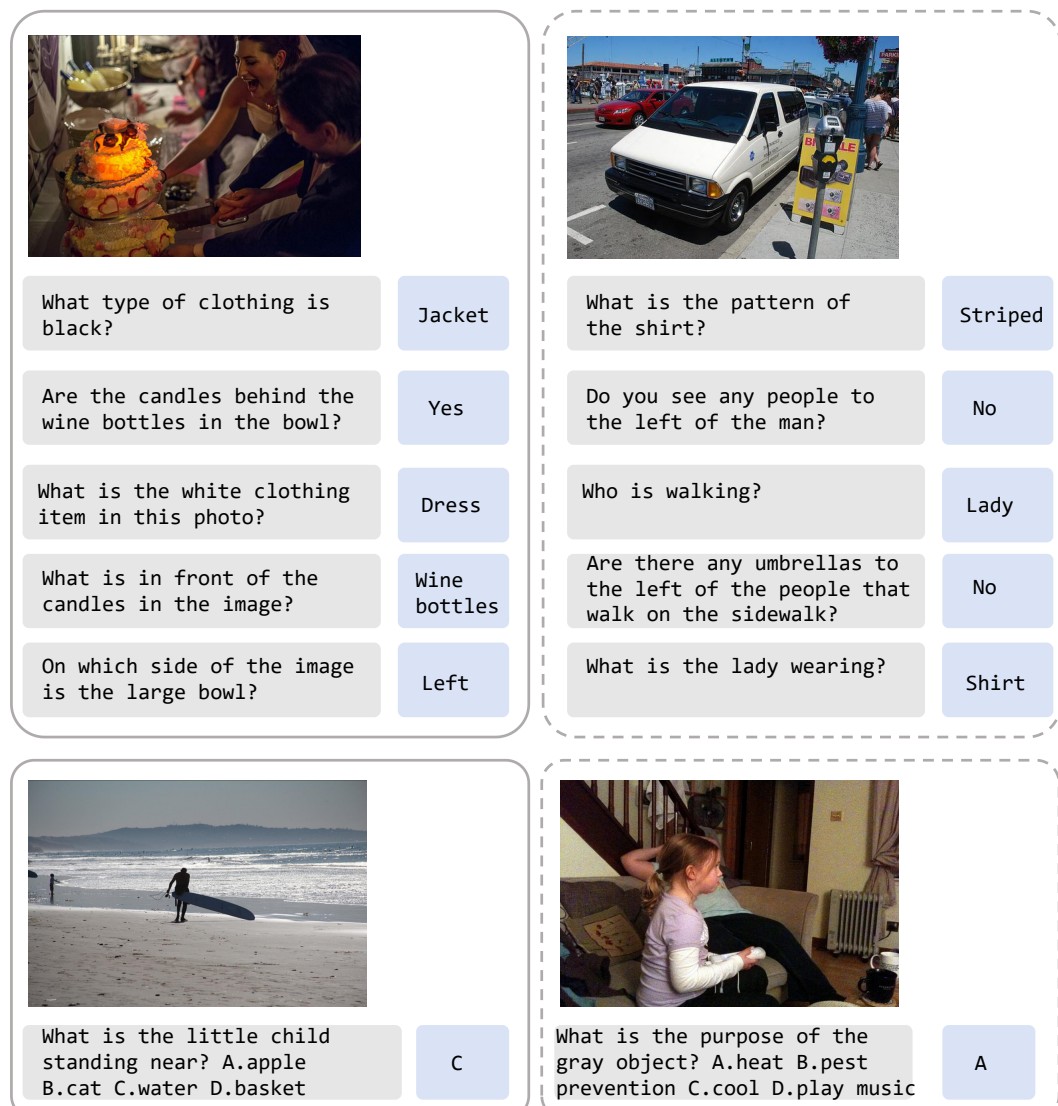

Figure 8: The best (top left) and the worst (top right) CoIDO scores samples from GQA datasets. The best (bottom left) and the worst (bottom right) CoIDO scores samples from VQAv2 datasets.

