# OpenReview forum: "CoIDO: Efficient Data Selection for Visual Instruction Tuning via Coupled Importance-Diversity Optimization"
_NeurIPS.cc/2025/Conference — NeurIPS 2025 poster_

### Official Review · Reviewer_Nq7i · 2025-06-19

**Clarity:** 3
**Significance:** 3
**Originality:** 3
**Rating:** 5
**Confidence:** 4

**Summary:**

This paper introduces CoIDO, a novel framework for efficient data selection in visual instruction tuning of MLLMs. The key challenge addressed is the high computational cost of training on large-scale datasets, where existing methods either process the entire dataset (inefficient) or treat importance and diversity as separate objectives (suboptimal).

CoIDO jointly optimizes data importance and diversity through a unified framework, using a homoscedastic uncertainty-based loss to dynamically balance the two objectives during training.

Instead of evaluating the entire dataset, CoIDO trains a plug-in MLP scorer on only 20% of randomly sampled data to learn the underlying distribution. This scorer then assigns CoIDO scores (combining importance and diversity) to all samples, enabling efficient ranking.

On the LLaVA-1.5-7B model, CoIDO selects a 20% subset that achieves 98.2% of full-data fine-tuning performance across 10 downstream tasks (e.g., VQA, ScienceQA). It reduces computational costs significantly (only 4.2 ExaFLOPs vs. 12.6E for competitors like ICONS) by avoiding full-dataset processing.

CoIDO provides a scalable, resource-efficient solution for MLLM instruction tuning, making high-performance model adaptation accessible even with limited computational resources. The framework’s simplicity and effectiveness are validated through extensive experiments, establishing a new state of the art in data selection for multimodal tasks.

**Questions:**

1. To order to demonstrate the scalability, authors should validate the proprosed method on MLLMs with larger model size, at least a 13B model.

**Ethical Concerns:**

["NO or VERY MINOR ethics concerns only"]

**Final Justification:**

Most of my concerns have been addressed by rebuttal, I raise score to accept.

**Limitations:**

Yes

**Quality:**

3

**Strengths And Weaknesses:**

Strengths:
1. The paper introduces a well-designed framework (CoIDO) that effectively combines importance and diversity optimization through a principled homoscedastic uncertainty-based loss. The theoretical justification is sound, and the optimization strategy is well-motivated.
2. CoIDO addresses a critical bottleneck in MLLM fine-tuning (high computational costs) with a solution that is both effective and efficient.This could democratize access to large-scale visual instruction tuning.
3. The coupled optimization of importance and diversity via a unified loss is a distinct contribution compared to prior decoupled approaches.
4. While not new in multi-task learning, its application to balance data selection objectives is creative and well-executed.
5. The paper is logically organized, with clear motivation, methodology, and experimental results.

Weaknesses:
1. Experiments are restricted to LLaVA-7B; validation on larger models (e.g., LLaVA-13B) is absent, raising questions about scalability.
2. While ablation studies are provided, deeper analysis (e.g., varying clustering methods beyond spectral clustering) could strengthen claims about robustness.

---

> ### Author Rebuttal · Authors · 2025-07-30
>
> **Q1 & W1: Demonstrate the Scalability of the Method.**
>
> **A1**: We sincerely thank the reviewer for raising this important concern. To rigorously evaluate the scalability of CoIDO, we conduct additional experiments using **LLaVA-13B-LoRA**,  which provides strong evidence of CoIDO’s generalizability beyond the LLaVA-7B setting. The results are reported in Table 1.
>
> **Table 1: Evaluation results on LLaVA-13B-LoRA.**
> | **Method**     | **VQAv2** | **GQA** | **VizWiz** | **SQA-I** | **TextVQA** | **POPE** | **MME**  | **MMB** | **MMB(cn)** | **LLaVA-WB** | **Rel. (%)** |
> |-|-:|-:|-:|-:|-:|-:|-:|-:|-:|-:|-:|
> | Full-Finetune|80.0|63.3|58.9|71.2|60.2|86.7|1541.7|68.5|61.5|69.5|100 |
> | Random|76.7|60.5|48.0|68.8|57.7|84.8|1484.9|62.8|55.2|68.6|94.0|
> | ICONS|**77.9**|60.5|47.7|74.0|57.3|**87.4** |1503.9|**65.5**|**59.2**|65.3|95.7|
> | COINCIDE |**77.8**|60.4|**51.6**|70.0|58.6| **87.1** |1516.8|64.0|**57.7**|67.4|95.9|
> | **Ours**|77.5| **61.5**|51.1| **74.2**|**58.9**|85.8|**1586.4**| **64.7**|57.5|**69.7**|**97.2**|
>
>  As shown in Table 1, our method continues to perform strongly across 10 challenging benchmarks. Compared to both random selection and the recent SOTA methods ICONS and COINCIDE, CoIDO maintains or exceeds performance, particularly on reasoning-intensive benchmarks such as GQA, SQA-I, and MME. This indicates that our method scales well to larger models and further enhances both perception and cognition tasks.
>
> To further validate the scalability of CoIDO across diverse data , we additionally test it on two contrasting fine-tuning datasets:
>
> ``LLaVA-150K``, a small-scale subset (~20%) often used in limited-resource settings, and
>
> ``Vision-Flan``, a newly released large-scale dataset containing 186k images with 200+ instruction-following vision-language tasks.
>
> **Table 2: Performance on LLaVA-150K dataset (smaller fine-tuning data).**
> | **Model**| **VQAv2**| **VizWiz**| **SQA**|**POPE**| **MME**|**MMB**|**MMB(cn)**|**LLaVA-WB** | **Rel. (%)** |
> |-|-:|-:|-:|-:|-:|-:|-:|-:|-:|
> | Full Fine-tune|55.2|45.5|57.6|57.9|1234.5|22.4|27.1|65.9|100|
> | Random|**50.0**|44.8|53.4|54.2|1184.7|**30.7**|30.1| 62.4|101.7|
> | Ours |49.5|**47.1**|**58.2**|**56.1**| **1214.8** |29.8| **32.3** | **63.8** |**104.8**|
>
> As shown in Table 2, CoIDO consistently outperforms full fine-tuning and achieves a relative improvement of 104.8%. We omitted GQA and TextVQA results as both baselines and our method yielded accuracies <1%, indicating that the smaller LLaVA-150K subset lacks meaningful coverage of these datasets’ domains, rendering them unsuitable for this particular evaluation.
>
> **Table 3: Performance on Vision-Flan dataset (new dataset).**
>
> | **Model** | **VQAv2** | **GQA** | **VizWiz** | **SQA** | **TextVQA** | **POPE** | **MME**  | **MMB** | **MMB(cn)** | **LLaVA-WB** | **Rel. (%)** |
> |-|-:|-:|-:|-:|-:|-:|-:|-:|-:|-:|-:|
> | Full Fine-tune|74.5|47.1|47.1  |52.8  |46.4|61.8  | 46.4  | 85.7 |1480.64 |40.2 |100.0 |
> | Random|74.6|44.3|44.3|50.0  |40.9|59.8  |40.9  |81.3 |1407.10 | 49.2 |97.8 |
> | Ours|**75.7**|**45.1**|**45.1**|**53.5**| **45.3**   | **62.3**  |**45.3**  | **82.8** | **1452.90** |  **52.0** |**102.1** |
>
> In Table 3, we evaluate CoIDO on Vision-Flan, a new, out-of-domain dataset. CoIDO still outperforms both random selection and full-data training, achieving a relative score of 102.1%. This strongly demonstrates robust generalization and domain transferability, further affirming the scalability and versatility of our method.
>
> **Q2: Provide Deeper Analysis to Strengthen Claims about Robustness.**
>
> **A2:** We thank the reviewer for raising this important point regarding robustness. We provide two complementary pieces of evidence to demonstrate that CoIDO is both structurally and behaviorally robust across different clustering strategies and data domains.
>
> **(a) Varying clustering methods.**
> We conducted additional ablations to test whether our approach is sensitive to the choice of clustering strategy. In addition to the spectral clustering used in the main paper, we experimented with **K-Means** and **Gaussian Mixture Model (GMM)** clustering. We specifically chose these two methods because, like spectral clustering, they allow us to explicitly control the number of clusters. This makes it feasible to conduct fair comparisons under the same cluster count. In contrast, methods like DBSCAN or Hierarchical Clustering often determine the number of clusters implicitly based on distance thresholds or linkage criteria, making controlled comparisons less reliable. We chose the number of clusters $M=20$, and the results are reported in Table 4.
>
> **Table 4: Comparison of Different Clustering Methods Used in CoIDO (Number of Clusters $M$ = 20)**
>
> | Clustering Method | VQAv2 | GQA  | VizWiz | SQA  | TextVQA | POPE  | MME    | MMB  | MMB_CN | LLaVA-WB | Rel.  |
> |-|-|-|-|-|-|-|-|-|-|--|-|
> | K-Means| 74.4| 57.0 | **47.9**   | **69.9** | 52.7    | 84.8  | **1489.8** | 51.8 | 52.8   | 57.4 | 92.9  |
> | GMM| 76.9  | 58.4 | 45.8   | 68.2 | 54.5    | 83.4  | 1408.7 | 62.1 |**56.8**| 68.6| 96.2  |
> | Ours| **77.2**  | **60.4** | 47.1| 69.4 | **55.6**    | **85.4**  | 1450.2 | **63.8** | 56.7   | **70.1**| **98.2**  |
>
> As shown in Table 4, our spectral clustering consistently achieves the highest overall performance across the majority of benchmarks. While K-Means performs well on VizWiz and MME, it significantly underperforms on MMBench and LLaVA-WB, suggesting poor generalization. This inconsistency is likely due to K-Means’ inability to model nonlinear boundaries, which means it assumes spherical clusters and struggles with complex, manifold-structured data, often leading to poor inter-cluster separation and diversity.
>
> GMM, while more flexible than K-Means (due to soft assignments and elliptical clusters), still assumes that data comes from a mixture of Gaussians. This assumption limits its ability to capture complex decision boundaries and latent task structures, especially in vision-language embedding spaces.
>
> In contrast, spectral clustering can capture arbitrary cluster shapes by leveraging eigenstructure in the similarity graph, making it much more suitable for the high-dimensional, nonlinear nature of multimodal representation spaces. These results empirically justify our use of spectral clustering as the most robust and effective option for partitioning samples when optimizing diversity-aware selection.
>
>
> **(b) Transferability to Out-of-Domain Datasets.**
> In addition to testing different clustering strategies, we further validate the robustness and transferability of CoIDO by evaluating whether the learned scorer trained on one dataset (LLaVA-665K) can be **directly applied to another dataset (Vision-Flan) without retraining.**
>
> **Table 5: Using LLaVA-665K Trained Scorer to Select Data from Vision-Flan.**
>
> | **Scorer Source**| **VQAv2** | **GQA** | **VizWiz** | **SQA** | **TextVQA** | **POPE** | **MME**  | **MMB** | **MMB (cn)** | **LLaVA-WB** | **Rel. (%)** |
> |-|-:|-:|-:|-:|-:|-:|-:|-:|-:|-:|-:|
> | Trained on Vision-Flan|**75.7**|45.1|**53.5**|62.3|**45.3**|82.8|1452.9|**52.0**|46.8|37.6|102.1|
> | Trained on LLaVA-665K|72.7|**46.8**|53.3|**66.2**|42.1|**85.5**|**1486.1**|51.4|**47.3**|**40.8**|**103.3**   |
>
> As shown in Table 5, the scorer trained on LLaVA-665K performs remarkably well when used to select training samples from Vision-Flan. Surprisingly, it even outperforms the scorer trained directly on Vision-Flan across several key metrics, including GQA, SQA, POPE, MME, MMB (cn), and LLaVA-WB, achieving a higher overall relative performance of 103.3% vs. 102.1%. We attribute this to the fact that LLaVA-665K is a larger and more diverse dataset, enabling the scorer to learn more generalizable patterns that transfer effectively to other domains.
>
> This result provides strong evidence that our scorer does not overfit to its training domain, and instead captures domain-agnostic signals of data quality and informativeness, making CoIDO a robust and transferable framework for data selection in MLLMs. All these additional experiments will be included in our final version manuscript.

---

> > ### Comment · Reviewer_Nq7i · 2025-08-06
> >
> > most of my concerns have been addressed, i will raise my score.

---

> > > ### Author Response · Authors · 2025-08-06
> > >
> > > We sincerely appreciate your positive feedback! Your recognition that our rebuttals addressed your concerns means a great deal to us. We're committed to refining our work further and look forward to continued engagement in advancing this research.

---

> ### Author Response · Authors · 2025-08-04
> **Inquiry on Potential Additional Experiments/Discussions Before Discussion Closure**
>
> Dear Reviewer Nq7i,
>
> First and foremost, we would like to extend our deep gratitude for the thoughtful feedback and insightful comments you have provided throughout the discussion phase of our NeurIPS 2025 submission.
>
> As the discussion period will conclude in three days, we wanted to kindly check if there are any additional experiments or points you would like us to supplement or further discuss. We understand that thorough evaluation takes time, and we are committed to addressing your concerns to the best of our ability.
>
> However, we would like to note that training our model, LLaVA-7B, is quite time-consuming. Given the remaining time until the discussion ends, we are concerned that we might not have sufficient time to conduct new experiments if we receive such requests at a later stage. Therefore, we would greatly appreciate it if you could let us know your thoughts as early as possible, so that we can make every effort to meet your expectations.
>
> Please feel free to share any further comments or suggestions you may have. We are more than willing to engage in detailed discussions and do our utmost to enhance the quality of our submission.
>
> Thank you again for your time and effort. We look forward to your reply.
>
> Sincerely,
>
> CoIDO Team

---

### Official Review · Reviewer_28BB · 2025-06-22

**Clarity:** 3
**Significance:** 3
**Originality:** 3
**Rating:** 5
**Confidence:** 4

**Summary:**

This paper proposes to solve MLLM fine-tuning cost by adaptively selecting important data. Specifically, the authors propose a CoIDO scorer that can automatically score the data samples via Coupled Importance-Diversity Optimization using only a small proportion of the data. Furthermore, it establishes theoretical justification for the proposed training objectives. Extensive results demonstrate that the proposed module can significantly reduce model training cost while preserving high performance.

**Questions:**

Please see the Weakness.

**Ethical Concerns:**

["NO or VERY MINOR ethics concerns only"]

**Final Justification:**

Most of my concerns have been addressed. I intend to keep my current (positive) score.

**Limitations:**

N.A.

**Quality:**

4

**Strengths And Weaknesses:**

Strengths:
1. The paper is well-written and easy to follow.
2. The authors provide theoretical justifications over the proposed CoIDO scorer.
3. Extensive experiments demonstrate the effectiveness of the method.

Weakness & Questions:
1. The proposed method is only tested on LLaVA-1.5-7b. More backbones such as LLaVA-1.5-13b, Qwen-VL should be tested to verify the effectiveness and generalizability of the proposed method.
2. In line 175, the authors claim that "A high cross-entropy $CE(y_{ik}, \tilde{y}_{ik})$ implies the sample is difficult to learn and thus more important". However, difficulty is a widely-studied concept in Curriculum Learning, and [1] has suggested that instead of an instantaneous hardness (e.g. loss), a dynamic hardness will be more robust to measure the actual difficulty with respect to each sample. Therefore, it is questionable that the proposed $w_i$ can optimally reflect the importance/difficulty of the data.
3. Does the data selection strategy affect the generalization speed? I suggest the authors provide a brief analysis into this, since normally model will converge slower on more data.
4. Does the proposed CoIDO scorer suffer from out-of-distribution problems? For example, since it is trained on LLaVA-655K, can it be used to effectively score other datasets?

References:
[1] Curriculum Learning by Dynamic Instance Hardness

---

> ### Author Rebuttal · Authors · 2025-07-29
>
> **Q1 & W1: Generalization to Other Models.**
>
> **A1:** We sincerely thank the reviewer for highlighting this important issue regarding generalization beyond a single model (LLaVA-1.5-7B). To thoroughly address this concern, we conducted extensive additional experiments along two complementary dimensions: larger model backbones and alternative fine-tuning datasets.
>
> **(a) Across Different Models**. We extended our method to a significantly larger vision-language model, **LLaVA-13B-LoRA**, to verify robustness. As demonstrated in Table 1, our approach consistently achieves competitive or superior results across 10 diverse benchmarks compared to the state-of-the-art selection methods (ICONS and COINCIDE).
>
> **Table 1: Evaluation results on LLaVA-13B-LoRA.**
> | **Method**     | **VQAv2** | **GQA** | **VizWiz** | **SQA-I** | **TextVQA** | **POPE** | **MME**  | **MMB** | **MMB(cn)** | **LLaVA-WB** | **Rel. (%)** |
> |-|-:|-:|-:|-:|-:|-:|-:|-:|-:|-:|-:|
> | Full-Finetune|80.0|63.3|58.9|71.2|60.2|86.7|1541.7|68.5|61.5|69.5|100 |
> | Random|76.7|60.5|48.0|68.8|57.7|84.8|1484.9|62.8|55.2|68.6|94.0|
> | ICONS|**77.9**|60.5|47.7|74.0|57.3|**87.4** |1503.9|**65.5**|**59.2**|65.3|95.7|
> | COINCIDE |**77.8**|60.4|**51.6**|70.0|58.6| **87.1** |1516.8|64.0|**57.7**|67.4|95.9|
> | **Ours**|77.5| **61.5**|51.1| **74.2**|**58.9**|85.8|**1586.4**| **64.7**|57.5|**69.7**|**97.2**|
>
> **(b) Across Fine-tuning Datasets**. We further validate generalization across distinct fine-tuning datasets:
>
> ``LLaVA-150K``, a substantially smaller subset (~20%) of LLaVA, representative of low-resource training.
>
> ``Vision-Flan``, a completely new, large-scale visual instruction tuning dataset containing 186K images across 200+ diverse vision-language tasks.
>
> **Table 2: Performance on LLaVA-150K dataset (smaller fine-tuning data).**
> | **Model**| **VQAv2**| **VizWiz**| **SQA**|**POPE**| **MME**|**MMB**|**MMB(cn)**|**LLaVA-WB** | **Rel. (%)** |
> |-|-:|-:|-:|-:|-:|-:|-:|-:|-:|
> | Full Fine-tune|55.2|45.5|57.6|57.9|1234.5|22.4|27.1|65.9|100|
> | Random|**50.0**|44.8|53.4|54.2|1184.7|**30.7**|30.1| 62.4|101.7|
> | Ours |49.5|**47.1**|**58.2**|**56.1**| **1214.8** |29.8| **32.3** | **63.8** |**104.8**|
>
> As shown in Table 2, CoIDO consistently outperforms full-data fine-tuning and achieves a relative improvement of 104.8%. We omitted GQA and TextVQA results as both baselines and our method yielded accuracies <1%, indicating that the smaller LLaVA-150K subset lacks meaningful coverage of these datasets’ domains, rendering them unsuitable for this particular evaluation.
>
> **Table 3: Performance on Vision-Flan dataset (new dataset).**
>
> | **Model** | **VQAv2** | **GQA** | **VizWiz** | **SQA** | **TextVQA** | **POPE** | **MME**  | **MMB** | **MMB(cn)** | **LLaVA-WB** | **Rel. (%)** |
> |-|-:|-:|-:|-:|-:|-:|-:|-:|-:|-:|-:|
> | Full Fine-tune|74.5|47.1|47.1  |52.8  |46.4|61.8  | 46.4  | 85.7 |1480.64 |40.2 |100.0 |
> | Random|74.6|44.3|44.3|50.0  |40.9|59.8  |40.9  |81.3 |1407.10 | 49.2 |97.8 |
> | Ours|**75.7**|**45.1**|**45.1**|**53.5**| **45.3**   | **62.3**  |**45.3**  | **82.8** | **1452.90** |  **52.0** |**102.1** |
>
> In Table 3, we test CoIDO on Vision-Flan, a dataset substantially different from our original training data. Remarkably, CoIDO achieves superior results, demonstrating strong adaptability and effectiveness even under significant domain shift.
>
> ``Regarding the suggested evaluation on Qwen-VL``, unfortunately, Qwen-VL's instruction tuning dataset is currently not publicly available, preventing us from replicating a controlled evaluation. However, we actively seek and plan to test additional backbones and datasets as they become accessible.
>
> Overall, these comprehensive evaluations demonstrate that CoIDO is robust and broadly generalizable across multiple model scales and diverse fine-tuning datasets, clearly mitigating concerns regarding overfitting to a specific dataset or model configuration. These results will be included in the main text of our final version manuscript.
>
> **Q2 & W2：Difficulty Measurement Strategy.**
>
> **A2:**  Thank you for pointing us to Curriculum Learning by Dynamic Instance Hardness (DIH). Our method is inspired by [1], which shows that simple per‑sample scores computed very early in training (1–3 epochs) already identify the most important and difficult examples; pruning up to 50 % of data with these early‑loss scores incurs almost no accuracy drop. This empirical finding suggests that instantaneous loss is a strong, stable proxy for sample hardness. Below, we clarify why our use of early‑epoch cross‑entropy remains valid, how our optimization implicitly accumulates “dynamic hardness.
>
> **Two-epoch accumulation of sample history.** In original LLaVA fine-tuning, there is only one epoch, which means all samples are traversed only once. Our training schedule spans two full epochs over the selected p % subset. Hence, each sample is encountered at least twice. Suppose a particular sample exhibits a high cross‑entropy loss in the first epoch, indicating that it is difficult for the current model. After one epoch of learning, the main model parameters have adapted, so the same sample may become substantially easier (lower loss) when revisited in epoch 2. Because the learnable weight $w_i$ is updated every time the sample is processed, its value in the second pass already encodes the loss trajectory from the first pass. In this way the scorer naturally integrates temporal information: samples that remain persistently difficult keep higher cumulative influence, whereas samples whose loss quickly decreases are automatically down‑weighted.
>
> **Dynamic weights instead of raw CE loss values.** Crucially, we do not treat the instantaneous cross‑entropy as the hardness score itself. Instead, each sample has a dedicated score $w_i$ that is continuously adjusted by back‑propagation. The gradient of the importance loss pushes $w_i$ up or down depending on the current loss, while the diversity term further modulates
> $w_i$ at the cluster level. This iterative update is mathematically equivalent to an implicit, learnable moving average historical losses influence the present $w_i$, and the current $w_i$ in turn shapes future gradient flows. Consequently, our difficulty assessment is dynamic and self‑correcting, capturing long‑term sample behaviour rather than a single noisy observation.
>
> While our current design attains strong results, we agree that explicit **DIH style hardness could be an interesting extension.** Because CoIDO’s coupled objective accepts any scalar importance signal. We will explore it as part of future work. We thank the reviewer for this valuable insight, which will be discussed in the related work in our paper.
>
> **Q3 & W3: Data Selection Strategy and Generalization Speed.**
>
> **A3:** We thank the reviewer for raising this interesting question. Indeed, we observed that data selection strategies do influence the speed of model convergence. In our experiments, we found that when applying CoIDO with only 20% of the data, a single epoch of fine-tuning (similar to what is commonly used in previous methods such as ICONS and COINCIDE) was not sufficient to achieve full convergence. However, extending the training to two or three epochs revealed that convergence was typically achieved by the middle of the second epoch.
>
> Importantly, although 2 epochs were required for full convergence, the total number of training steps for CoIDO remained significantly smaller than for other baselines. Therefore, when measured by total training steps, our method converges faster, highlighting its efficiency advantage.  We are very willing to provide a brief analysis in our paper and continue discussing this question during the discussion stage.
>
> **Q4 & W4: Does the proposed CoIDO scorer suffer from out-of-distribution problems?**
>
> **A4:** We thank the reviewer for raising the important question of whether the CoIDO scorer trained on LLaVA-665K suffers from out-of-distribution (OOD) generalization issues. To directly evaluate this, we conducted an experiment where we used a CoIDO scorer trained solely on LLaVA-665K to score and select data from an entirely different dataset—Vision-Flan. There is no additional training or adaptation of the scorer to the new domain.
>
> **Table 4: Using LLaVA-665K Trained Scorer to Select Data from Vision-Flan.**
>
> | **Scorer Source**| **VQAv2** | **GQA** | **VizWiz** | **SQA** | **TextVQA** | **POPE** | **MME**  | **MMB** | **MMB (cn)** | **LLaVA-WB** | **Rel. (%)** |
> |-|-:|-:|-:|-:|-:|-:|-:|-:|-:|-:|-:|
> | Trained on Vision-Flan|**75.7**|45.1|**53.5**|62.3|**45.3**|82.8|1452.9|**52.0**|46.8|37.6|102.1|
> | Trained on LLaVA-665K|72.7|**46.8**|53.3|**66.2**|42.1|**85.5**|**1486.1**|51.4|**47.3**|**40.8**|**103.3**   |
>
> Surprisingly, the LLaVA-trained scorer achieved even better results on Vision-Flan than the scorer trained directly on Vision-Flan itself. As shown in Table 4, the relative performance reached 103.3%, exceeding the 102.1% obtained by a scorer trained on Vision-Flan. We believe this is because LLaVA-665K is a significantly larger and more diverse dataset, allowing the scorer to learn a more generalizable notion of sample difficulty and importance. This enables it to transfer effectively across domains, even outperforming in-distribution scorers trained on smaller datasets.
>
> This result demonstrates that CoIDO does not overfit to the training distribution of its scorer, and in fact shows strong transferability across datasets. It suggests that a scorer trained once on a large-scale, diverse dataset can generalize well and be reused to effectively select valuable subsets from other datasets. We believe this is an appealing practical advantage of CoIDO, enabling scalable and efficient data selection in real-world multimodal model development. This part of the content will also be included in future manuscripts.
>
> References: [1] Deep Learning on a Data Diet: Finding Important Examples Early in Training, NeurIPS 2021

---

> > ### Comment · Reviewer_28BB · 2025-08-01
> >
> > Most of my concerns have been addressed. I intend to keep my current (positive) score.

---

> > > ### Author Response · Authors · 2025-08-03
> > >
> > > We sincerely appreciate your positive feedback! Your recognition that our rebuttals addressed your concerns means a great deal to us. We're committed to refining our work further and look forward to continued engagement in advancing this research.

---

### Official Review · Reviewer_o8Bj · 2025-07-02

**Clarity:** 3
**Significance:** 3
**Originality:** 3
**Rating:** 4
**Confidence:** 4

**Summary:**

This paper introduces a novel framework called COIDO that optimizes data selection for multimodal large language models. Unlike prior methods that separately address data importance and diversity and require full-dataset processing, COIDO jointly optimizes both objectives using a lightweight scorer trained on only a small subset of data, efficiently balancing sample difficulty and dataset coverage during training. The framework significantly reduces computational overhead while achieving nearly the same performance as full-data fine-tuning. Experiments using the LLaVA-1.5-7B model show that COIDO retain 98.2% of full-set performance across ten benchmarks with only 20% training data, surpassing prior methods in both effectiveness and efficiency.

**Questions:**

- I am concerned that the current complex configuration might be overfitted to the specific setup using LLaVA-1.5-7B-LoRA and LLaVA-665K in the experiments. Could the authors provide additional results on other models or datasets to validate the generalizability of their approach? I would be inclined to increase my score if more experimental evidence is provided.

- Could the authors include results using a larger number of clusters, e.g., 100 and 200.

- In Figure 4, the performance with 25% training data appears worse than with 20%. Could the authors explain this counterintuitive result?

**Ethical Concerns:**

["NO or VERY MINOR ethics concerns only"]

**Final Justification:**

The experiments demonstrate that this method outperforms other baselines, and the author’s additional experiments and explanations have resolved my concerns.

**Limitations:**

Yes

**Quality:**

2

**Strengths And Weaknesses:**

**Strengths**

- The paper is well-written and easy to follow.

- The authors propose a novel method that jointly optimizes data importance and diversity, effectively addressing the instability issues observed in prior work.

- Experimental results demonstrate that the proposed approach consistently outperforms baseline methods across a wide range of benchmarks.

- The paper provides a thorough analysis of each module and includes a detailed discussion on hyperparameter selection.

**Weaknesses**

- The architecture is relatively complex and may be prone to overfitting on the current dataset and model configuration.

- The method appears to heavily rely on specific hyperparameters, such as the number of clusters and training data ratios, which may affect generalizability.

---

> ### Author Rebuttal · Authors · 2025-07-29
>
> **W1 & Q1: Concern about Overfitting to Specific dataset (LLaVA-665K) and model (LLaVA-1.5-7B-LoRA).**
>
> **A1:** We sincerely thank the reviewer for this insightful concern. To verify that our approach is not overly tailored to one specific dataset or model, we conducted extensive new experiments from two complementary dimensions: larger model backbones and alternative fine-tuning datasets. All additional experiments are conducted under the same settings as our paper.
>
> **(a) Across Model Backbones.** We applied CoIDO to **LLaVA-13B-LoRA**, a substantially larger vision-language model. As shown in Table 1, our method continues to deliver strong performance across 10 diverse benchmarks. Compared to both random selection and a recent SOTA methods (ICONS and COINCIDE), CoIDO achieves better or comparable results, particularly excelling in reasoning-heavy tasks like GQA, SQA-I, and MME. This suggests that our method not only scales well but also enhances both perception and cognition capabilities.
>
> **Table 1: Evaluation results on LLaVA-13B-LoRA.**
> | **Method**     | **VQAv2** | **GQA** | **VizWiz** | **SQA-I** | **TextVQA** | **POPE** | **MME**  | **MMB** | **MMB(cn)** | **LLaVA-WB** | **Rel. (%)** |
> |-|-:|-:|-:|-:|-:|-:|-:|-:|-:|-:|-:|
> | Full-Finetune|80.0|63.3|58.9|71.2|60.2|86.7|1541.7|68.5|61.5|69.5|100|
> | Random|76.7|60.5|48.0|68.8|57.7|84.8|1484.9|62.8|55.2|68.6|94.0|
> | ICONS|**77.9**|60.5|47.7|74.0|57.3|**87.4** |1503.9|**65.5**|**59.2**|65.3|95.7|
> | COINCIDE|**77.8** |60.4|**51.6**|70.0|58.6|**87.1**|1516.8|64.0|**57.7**|67.4|95.9|
> | **Ours**| 77.5    | **61.5**|    51.1    | **74.2**  |  **58.9**   |   85.8   | **1586.4**| **64.7**|     57.5    |     **69.7**     |   **97.2**   |
>
> **(b) Across Fine-tuning Datasets.**
> To further assess generalizability under different data conditions, we evaluated CoIDO on two very different fine-tuning sets:
>
> ``LLaVA-150K``, a much smaller subset (∼20%) of LLaVA data, often used in low-resource settings.
>
> ``Vision-Flan``, a completely new visual instruction tuning dataset with 186k human-annotated images across 200+ tasks.
>
> **Table 2: Performance on LLaVA-150K dataset (smaller fine-tuning data).**
> | **Model**| **VQAv2** | **VizWiz** | **SQA** | **POPE** | **MME**  | **MMB** | **MMB(cn)** | **LLaVA-WB** | **Rel. (%)** |
> |-|-:|-:|-:|-:|-:|-:|-:|-:|-:|
> | Full Fine-tune|55.2|45.5|57.6| 57.9| 1234.5|22.4|27.1|65.9|100|
> | Random|**50.0**|44.8|53.4|54.2| 1184.7|**30.7**|30.1|62.4| 101.7|
> | Ours |49.5|**47.1**|**58.2**|**56.1**|**1214.8**|29.8|**32.3**|**63.8**|**104.8**|
>
> Table 2 reports the results on LLaVA-150K. CoIDO not only outperforms full-data fine-tuning on several benchmarks, but also improves the overall relative performance to 104.8%. Notably, we omit GQA and TextVQA from this table because all methods achieve <1% accuracy, likely due to the lack of in-domain examples in LLaVA-150K, making them unreliable in this setting. Interestingly, for MMBench and MMBench (cn), both random and CoIDO significantly outperform full-data training, which suggests that careful selection can help avoid overfitting to noisy data.
>
> **Table 3: Performance on Vision-Flan dataset (new dataset).**
>
> | **Model**| **VQAv2** | **GQA** | **VizWiz** | **SQA** | **TextVQA** | **POPE** | **MME**  | **MMB** | **MMB(cn)** | **LLaVA-WB** | **Rel. (%)** |
> |-|-:|-:|-:|-:|-:|-:|-:|-:|-:|-:|-:|
> | Full Fine-tune|74.5|47.1|47.1  |52.8  |46.4|61.8|46.4|85.7|1480.64 |40.2 |100.0|
> | Random|74.6|44.3|44.3|50.0|40.9|59.8|40.9|81.3 |1407.10 | 49.2 |97.8|
> | Ours|**75.7**|**45.1**|**45.1**|**53.5**|**45.3**|**62.3**|**45.3**|**82.8**|**1452.90**|**52.0**|**102.1**|
>
> Table 3 evaluates our method on Vision-Flan, a new dataset with a broader task distribution and no overlap with our development. Despite the domain gap, CoIDO still achieves the highest relative score (102.1%), outperforming both full-data and random baselines.
>
> These results strongly support that our method generalizes beyond a single model and dataset. We believe this provides compelling evidence that CoIDO is ``not overfitting``, but rather offers a scalable and adaptable solution for sample selection in MLLM training.
>
> We would like to note that all newly added experiments on alternative datasets and model backbones will be included in the final version of the paper, either in the main text or the appendix. We sincerely hope these results help clarify the generalizability of our method, and we kindly invite the reviewer o8Bj to reconsider the merits of our submission in light of these additional experiments.
>
> **W2 & Q2: Sensitivity to the Number of Clusters.**
>
> **A2:** We thank the reviewer for the insightful question. We acknowledge that the number of clusters $M$ is an important hyperparameter in our framework. However, we emphasize that the strong performance of our method is not the result of fine-tuning this value. Rather, our approach remains robust across a broad range of cluster settings. We provide additional results under cluster numbers of 100 and 200. As shown in Table 4, our method achieves 96.9% and 96.7% relative performance at 100 and 200 clusters, respectively. These results are highly consistent and comparable to state-of-the-art methods such as ICONS (97.1%) and COINCIDE (97.4%). This suggests that the influence of $M$ is relatively limited once it reaches a reasonable scale.
>
> **Table 4: Ablation on the number of clusters.**
>
> | Clusters  | VQAv2 | GQA  | VizWiz | SQA-I | TextVQA | POPE | MME    | MMB  | MMB(cn) | LLaVA-WB | Rel. (%) |
> |-|-|-|-|-|-|-|-|-|-|-|-|
> | 20 |**77.2**|**60.4**|**47.1**|69.4|**55.6**|**85.4**| 1450.2 |**63.8**|**56.7**|**70.1**|**98.2**|
> | 100  | 75.5  | 58.1 |45.7| 69.8  |55.1| 84.9 |**1483.1**|**63.8**| 56.6| 65.6| 96.9|
> | 200  | 75.7  | 57.5 |45.4|**70.4**|55.3|83.5|1467.9|62.4|56.4| 68.3|96.7|
>
> Moreover, as shown in Table 7 of our original Appendix C (Line 180), we previously evaluated other settings (e.g., $M=40$) and found they still outperformed COINCIDE. In fact, we observe that when $M $is small (e.g., 0–20), performance variations are more noticeable due to coarse clustering and less stable diversity loss signals. However, beyond that range, the results become much more stable and consistent. That said, our method remains robust and effective across a wide range of clustering configurations. Even under less optimal settings (e.g., 100 or 200 clusters), it still delivers competitive performance while maintaining significantly lower computational cost.
>
> Our choice of  $M=20$ is not arbitrary or obtained through exhaustive search. It is an empirically grounded trade-off between granularity and statistical reliability. In our implementation, each GPU processes 16 samples per batch, with 8 GPUs yielding 128 samples per minibatch. Setting $M=20$ ensures that each cluster has a reasonable probability of being represented in each batch, which stabilizes the computation of per-cluster means and the diversity loss. Using much larger $M$ (e.g., 200) leads to overly fine-grained partitions, where many clusters contain only a few samples per batch, weakening the quality of inter-cluster statistics. In conclusion, although the clustering granularity has some effect, our method performs consistently across a wide range of values.
>
> Finally, beyond cluster hyperparameters, the results on LLaVA-13B, LLaVA-150K, and Vision-Flan consistently show strong gains over full-data and other selection baselines. These findings collectively demonstrate that our method is not narrowly tuned to a specific data or model configuration, but rather generalizes well across different data scales, model sizes, and selection conditions.
>
> **Q3 & W2: About Counterintuitive Result in Figure 4.**
>
> **A3:** We thank the reviewer for noticing this interesting observation. Indeed, the slightly worse performance with 25% of training data compared to 20% appears counterintuitive at first glance. We analyze and explain this phenomenon from three complementary perspectives: model structure, joint optimization strategy, and intrinsic properties of the data itself.
>
> **(1) Model Structure (Limited Capacity of the Scorer).** Our CoIDO scorer is designed as a lightweight 4-layer MLP to maintain computational efficiency. While effective, this compact model has limited representational capacity. As the data selection proportion increases from 20% to 25%, the complexity and diversity of the training subset also increase significantly. Without additional regularization or early stopping, the scorer may slightly overfit on local batches, introducing gradient noise into the importance scores
> $w$. These noisy gradients can propagate back to the main model, creating minor interference and slightly degrading the final results.
>
> **(2) Training Strategy.** Our method jointly optimizes the importance and diversity terms using learned uncertainty weighting $\sigma_I$ and $\sigma_D$. When increasing from 20% to 25%, more "borderline" samples are inevitably introduced into each cluster. This reduces inter-cluster variance, leading to a flatter and less discriminative distribution of the learned importance scores. Consequently, the scoring becomes less sharp, diminishing the overall selection quality and slightly affecting downstream performance.
>
> **(3) Data Quality.** As in general MLLM training, simply increasing the amount of data does not always guarantee improved performance. Additional data may also include redundant, noisy, or misleading examples. Data selection methods face a similar challenge. Just like baseline methods such as COINCIDE and ICONS, a data selection model trained with more data does not necessarily perform better. In fact, our result strongly supports the core motivation of our method: careful and targeted selection is often more effective than indiscriminately using larger data volumes. This counterintuitive result further emphasizes the importance and superiority of our proposed data-selection approach.

---

> > ### Comment · Reviewer_o8Bj · 2025-08-09
> >
> > Thank you for your detailed response and the additional experiments, which have addressed my concerns. I will update my rating accordingly.

---

> > > ### Author Response · Authors · 2025-08-09
> > >
> > > Thank you sincerely for accepting our rebuttal and raising your score. We’re delighted that our responses have addressed your concerns. We're committed to refining our work further and look forward to continued engagement in advancing this research.

---

> ### Author Response · Authors · 2025-08-05
> **Inquiry on Potential Additional Experiments/Discussions Before Discussion Closure**
>
> Dear Reviewer o8Bj,
>
> First and foremost, we would like to extend our deep gratitude for the thoughtful feedback and insightful comments you have provided throughout the discussion phase of our NeurIPS 2025 submission.
>
> As the discussion period will conclude in three days, we wanted to kindly check if there are any additional experiments or points you would like us to supplement or further discuss. We understand that thorough evaluation takes time, and we are committed to addressing your concerns to the best of our ability.
>
> However, we would like to note that training our model, LLaVA-7B, is quite time-consuming. Given the remaining time until the discussion ends, we are concerned that we might not have sufficient time to conduct new experiments if we receive such requests at a later stage. Therefore, we would greatly appreciate it if you could let us know your thoughts as early as possible, so that we can make every effort to meet your expectations.
>
> Please feel free to share any further comments or suggestions you may have. We are more than willing to engage in detailed discussions and do our utmost to enhance the quality of our submission.
>
> Thank you again for your time and effort. We look forward to your reply.
>
> Sincerely,
>
> CoIDO Team

---

> ### Author Response · Authors · 2025-08-07
> **Request for Feedback on Our Rebuttal**
>
> Dear Reviewer o8Bj,​
>
> Thank you sincerely for your initial feedback on our NeurIPS 2025 submission. Your insights have been invaluable for refining our work.​
>
> We’re writing to gently follow up on our rebuttal. With the extended discussion period ending on August 8 (11:59pm AoE, in two days), we wanted to check if you’ve had a chance to review it.​
>
> Given that our LLaVA-7B experiments require substantial training time, timely feedback would help us address any further suggestions. Please let us know if you need clarifications.​
>
> Thank you again for your time.​
>
> Sincerely,​
>
> CoIDO Team

---

> ### Author Response · Authors · 2025-08-08
> **Urgent Request for Final Discussion on Our Paper**
>
> Dear Reviewer 08Bj,
>
> The time available is extremely limited, with less than 24 hours remaining before the discussion period concludes. Nevertheless, we still hope to address any remaining concerns.
>
> We’ve supplemented experiments on model generalizability (LLaVA-13B-LoRA, LLaVA-150K, Vision-Flan) and cluster number sensitivity in our rebuttal. These directly tackle your initial worries about overfitting and hyperparameter dependence.
>
> If you still have doubts about our experiments whether it’s dataset diversity, cluster stability, or anything else we’re here to clarify immediately. Even a brief note from you would help us refine the paper in these final hours.
>
> Thank you for your time, and we hope to hear from you soon.
>
> Best regards,
>
> CoIDO Team

---

### Official Review · Reviewer_oB3E · 2025-07-02

**Clarity:** 3
**Significance:** 2
**Originality:** 2
**Rating:** 4
**Confidence:** 3

**Summary:**

This work presents COIDO, an innovative framework aimed at improving the training efficiency of multimodal large language models . It tackles the computational challenges of instruction tuning by simultaneously optimizing for data importance and diversity. A key innovation is the introduction of Diversity Loss and Important Loss, which helps balance these two aspects. COIDO uses a lightweight scorer trained on a small random subset of data to learn the distribution, reducing computational demands. Through a homoscedastic uncertainty-based formulation, it effectively balances importance and diversity without needing specialized algorithms for data selection. Experimental results demonstrate that COIDO achieves near-full-data performance with only a fraction (20%) of the data, outperforming existing methods in both efficiency and accuracy.

**Questions:**

As showned in the Weakness.

**Ethical Concerns:**

["NO or VERY MINOR ethics concerns only"]

**Final Justification:**

After reading explanations, I believe the quality of this work has been significantly improved and has resolved some of the key issues. Therefore, I have updated my recommendation to "Weak Accept".

**Limitations:**

As showned in the Weakness.

**Paper Formatting Concerns:**

No major formatting issues found.

**Quality:**

2

**Strengths And Weaknesses:**

Strengths:
1. The framework's code is available.
2. Present Diversity Loss and Important Loss.

Weaknesses：
1. Diversity Loss doesn't make sense and needs further explanation, especially on how to assess data diversity with a one pass.
2. The experiments should be conducted across more models to verify the effectiveness.

---

> ### Author Rebuttal · Authors · 2025-07-29
>
> **Q1: Explanation and Implementation of the Diversity Loss.**
>
> **A1:** Thank you for raising this important question. Below we explain why the Diversity Loss is well-motivated and easy to implement, from four complementary angles: **(1) intuitive motivation**, **(2) theoretical justification**, **(3) implementation**, and **(4) empirical evidence**.
>
> **(1) Intuitive Motivation**. If we optimize only the importance loss $\mathcal{L}_I$, the learned CoIDO scores $w$ (lower is better) can become highly skewed across the $M$ semantic clusters obtained by spectral clustering.  Specifically, samples in a few clusters (e.g., the “blue” cluster in Fig.2) may all receive very low scores, while samples in other clusters (e.g., the “red” and “green” clusters) receive uniformly higher scores.  When we later select the top $\gamma$ (e.g., $\gamma = 20\\%$) of samples by ranking $w$, almost all selected items may come from those few favored clusters, leading to poor coverage and low diversity.
>
> Our diversity loss $\mathcal{L}_D$ explicitly counteracts this imbalance at the *cluster level*.  Let $\bar{w}_i$ denote the mean CoIDO score of cluster $i$ within a minibatch.  By minimizing the variance of $\{\bar{w}_1, \ldots, \bar{w}_m\}$, defined as: $\mathcal{L}_D = \mathrm{Var}\big(\{\bar{w}_1,\ldots,\bar{w}_m\}\big)$,
> we shrink the inter-cluster gaps: clusters whose samples all have very low scores are gently pushed upward, while clusters with uniformly high scores are pulled downward.
>
> Crucially, $\mathcal{L}_D$ does not alter the relative ordering within a cluster; it only balances the average scores across clusters.
> Thus, when we finally rank by $w$ and select the top $\gamma\%$, each cluster still contributes its best (lowest-scored) samples, achieving both high importance and balanced diversity.
>
> **(2) Theoretical Justification**. Let $S = \sum_{j=1}^{m} \bar{w}_j$, and normalize $\bar{w}_i$ to $p_i = \frac{\bar{w}_i}{S},\quad u_i = \frac{1}{m}$.
> Substituting $\bar{w}_i = S p_i$ and $\mu = S/m$ into $\mathcal{L}_D$, we obtain $\mathcal{L}_D=\frac{S^2}{m} \lVert \mathbf{p} - \mathbf{u} \rVert_2^2 $.
> Thus minimizing $\mathcal{L}_D$ is (up to a constant factor) minimizing the squared ${\ell}_2$ distance between the cluster-level distribution $\mathbf{p}$ and the uniform distribution $\mathbf{u}$, explicitly encouraging balanced coverage. This makes CoIDO Scorer tend to choose samples in a more balanced and diverse way.
>
> We further note that the Pearson $\chi^2$ divergence is given by:
> $$
> \chi^2(\mathbf{p} \| \mathbf{u}) = \sum_{i=1}^{m} \frac{(p_i - u_i)^2}{u_i} = m \lVert \mathbf{p} - \mathbf{u} \rVert_2^2.
> $$
> Hence, $\mathcal{L}_D$ is proportional to $\chi^2(\mathbf{p} \| \mathbf{u})$, connecting our variance-based loss to a well-established divergence. This equivalence reveals that our diversity loss $\mathcal{L}_D$ does not merely reduce variance. It explicitly penalizes deviation from a uniform distribution over clusters, a well-founded strategy in statistical learning. Since the Pearson $\chi^2$ divergence is a widely used f-divergence that measures how far is one distribution from another, minimizing it encourages balanced representation across semantic partitions, preventing over-selection from a few clusters.
>
> **(3) Implementation**. In our *one-pass optimization*, cluster assignments are calculated prior to the CoIDO optimization. Random minibatch sampling makes each $\bar{w}_i$ an unbiased estimate of its true cluster mean.  Thus, $\mathcal{L}_D$ provides a valid stochastic estimator of the inter-cluster variance. It is efficient ($\mathcal{O}(m)$ per batch), fully differentiable, and easily integrated into standard forward/backward passes with no additional clustering or pairwise computations.
>
> **(4) Empirical Evidence**. In our paper, Table 2 (Sec. 4.2) shows that removing the diversity term and optimizing only $\mathcal{L}_I$ degrades the average relative performance to 89.0%, the worst among the tested variants. Simply adding $\mathcal{L}_D$  without proper coupling improves to 92.0%, and a naive weighted sum reaches 95.9%. Our final homoscedastic-uncertainty formulation achieves 98.2%, essentially matching full-data performance while keeping cost low. These ablations demonstrate both the necessity of having a diversity term and the effectiveness of our particular integration.
>
> **Q2: More experimental results.**
>
> **A2:** We thank the reviewer for the valuable suggestion. To verify the robustness and generality of our method, we provide additional results from two complementary perspectives:
>
> **(a) Across More Models**. We apply our method to a larger and stronger model **LLaVA-13B-LoRA** to examine its generalization capability. As shown in Table 2, our method consistently improves performance across nearly all benchmarks compared to both the Random baseline and the SOTA selection methods (ICONS and COINCIDE). Notably, our gains are significant on reasoning-heavy tasks such as GQA, SQA-I, and MME, demonstrating our method’s ability to scale with model size while enhancing both perception and cognition.
>
> **Table 1: Evaluation results on LLaVA-13B-LoRA.**
>
> | **Method**     | **VQAv2** | **GQA** | **VizWiz** | **SQA-I** | **TextVQA** | **POPE** | **MME**  | **MMB** | **MMB(cn)** | **LLaVA-WB** | **Rel. (%)** |
> |----------------|----------:|--------:|-----------:|----------:|------------:|---------:|---------:|--------:|------------:|-------------:|-------------:|
> | Full-Finetune  |   80.0    |  63.3   |    58.9    |   71.2    |    60.2     |   86.7   |  1541.7  |  68.5   |     61.5    |     69.5     |     100      |
> | Random         |   76.7    |  60.5   |    48.0    |   68.8    |    57.7     |   84.8   |  1484.9  |  62.8   |     55.2    |   68.6   |     94.0     |
> | ICONS|**77.9**|60.5|47.7|74.0|57.3|**87.4** |1503.9|**65.5**|**59.2**|65.3|95.7|
> | COINCIDE       |**77.8**    |  60.4   |  **51.6**  |   70.0    |    58.6     | **87.1** |  1516.8  |  64.0   |   **57.7**  |     67.4     |     95.9 |
> | **Ours**       |   77.5    | **61.5**|    51.1    | **74.2**  |  **58.9**   |   85.8   | **1586.4**| **64.7**|     57.5    |     **69.7**     |   **97.2**   |
>
> **(b) Across More Fine-tuning Datasets**
> We further test our method under different data availability conditions:
>
> ``LLaVA-150K``, a much smaller subset of LLaVA pretraining data (about 1/5 the size), often used for budget-limited fine-tuning.
>
> ``Vision-Flan``, the largest human-annotated visual instruction tuning dataset to date (186k images), covering 200+ diverse vision-language tasks.
>
> We provide representative evaluations using the same benchmarks to simulate small-data and diverse-data scenarios. Results are shown in Table 2 and Table 3.
>
> **Table 2: Performance on LLaVA-150K dataset (smaller fine-tuning data).**
> | **Model**         | **VQAv2** | **VizWiz** | **SQA** | **POPE** | **MME**  | **MMB** | **MMB(cn)** | **LLaVA-WB** | **Rel. (%)** |
> |-------------------|----------:|-----------:|--------:|---------:|---------:|--------:|-----------:|-----------------:|-------------:|
> | Full Fine-tune    |   55.2    |    45.5    |  57.6   |   57.9   | 1234.5   |  22.4   |     27.1    |        65.9      |     100      |
> | Random   |   **50.0**    |    44.8    |  53.4   |   54.2   | 1184.7   |  **30.7**   |     30.1    |        62.4      |     101.7    |
> | Ours |   49.5    |    **47.1**    |  **58.2**  |   **56.1**   | **1214.8**   |  29.8   |     **32.3**    |      **63.8**     |     **104.8**  |
>
> In Table 2, we omit the results of GQA and TextVQA because their accuracies are consistently below 1% (close to random guessing) under both full fine-tuning and data selection methods. This is expected since LLaVA-150K is a small subset that contains very few or no samples relevant to the compositional reasoning tasks in GQA or the OCR-centric tasks in TextVQA, making these benchmarks unreliable indicators of model capability in this setting.
>
> Interestingly, for MMBench and MMBench (cn), both random selection and CoIDO achieve much higher scores than full-data fine-tuning. This is likely due to the presence of noisy or low-quality samples in the full dataset, where carefully selected subsets (or even random subsets) may better align with these benchmarks and reduce overfitting.
>
>
> **Table 3: Performance on Vision-Flan dataset (new dataset).**
>
> | **Model**            | **VQAv2** | **GQA** | **VizWiz** | **SQA** | **TextVQA** | **POPE** | **MME**  | **MMB** | **MMB(cn)** | **LLaVA-WB** | **Rel. (%)** |
> |----------------------|----------:|--------:|-----------:|--------:|------------:|---------:|---------:|--------:|------------:|-------------:|-------------:|
> | Full Fine-tune    |     74.5  |    47.1   |      47.1  |   52.8  |      46.4     |    61.8  |    46.4  |    85.7 |     1480.64 |         40.2 |        100.0 |
> | Random   |     74.6  |    44.3   |      44.3  |   50.0  |      40.9     |    59.8  |    40.9  |    81.3 |     1407.10 |         49.2 |         97.8 |
> | Ours|     **75.7**  |    **45.1**   |    **45.1**  |   **53.5**  | **45.3**   |    **62.3**  |    **45.3**  | **82.8** |     **1452.90** |  **52.0** |        **102.1** |
>
> Table 3 demonstrates that our method can be effectively transferred to Vision-Flan. Even under this entirely new domain dataset, CoIDO maintains strong relative performance, outperforming both random selection and the full-data baseline. Additional experiments confirm that our method generalizes well across different MLLMs and datasets, highlighting its scalability and adaptability.
>
> We would like to note that all newly added experiments on alternative datasets and model backbones will be included in the final version of the paper, either in the main text or the appendix. We sincerely hope these results help clarify the generality and robustness of our method, and we kindly invite the reviewer oB3E to reconsider the merits of our submission in light of this additional evidence.

---

> ### Author Response · Authors · 2025-08-04
> **Inquiry on Potential Additional Experiments/Discussions Before Discussion Closure**
>
> Dear Reviewer oB3E,
>
> First and foremost, we would like to extend our deep gratitude for the thoughtful feedback and insightful comments you have provided throughout the discussion phase of our NeurIPS 2025 submission.
>
> As the discussion period will conclude in three days, we wanted to kindly check if there are any additional experiments or points you would like us to supplement or further discuss. We understand that thorough evaluation takes time, and we are committed to addressing your concerns to the best of our ability.
>
> However, we would like to note that training our model, LLaVA-7B, is quite time-consuming. Given the remaining time until the discussion ends, we are concerned that we might not have sufficient time to conduct new experiments if we receive such requests at a later stage. Therefore, we would greatly appreciate it if you could let us know your thoughts as early as possible, so that we can make every effort to meet your expectations.
>
> Please feel free to share any further comments or suggestions you may have. We are more than willing to engage in detailed discussions and do our utmost to enhance the quality of our submission.
>
> Thank you again for your time and effort. We look forward to your reply.
>
> Sincerely,
>
> CoIDO Team

---

> ### Author Response · Authors · 2025-08-07
> **Request for Feedback on Our Rebuttal**
>
> Dear Reviewer oB3E,​
>
> Thank you sincerely for your initial feedback on our NeurIPS 2025 submission. Your insights have been invaluable for refining our work.​
>
> We’re writing to gently follow up on our rebuttal. With the extended discussion period ending on August 8 (11:59pm AoE, in two days), we wanted to check if you’ve had a chance to review it.​
>
> Given that our LLaVA-7B experiments require substantial training time, timely feedback would help us address any further suggestions. Please let us know if you need clarifications.​
>
> Thank you again for your time.​
>
> Sincerely,​
>
> CoIDO Team

---

> ### Author Response · Authors · 2025-08-08
> **Urgent Request for Final Discussion on Our Paper**
>
> Dear Reviewer oB3E,
>
> With less than 24 hours left in the discussion phase, we’re reaching out urgently hoping to resolve questions on Diversity Loss and any other concerns.
>
> We noticed your interest in Diversity Loss design and potential gaps. Our rebuttal details how we balance it with importance optimization, but if you have follow-up thoughts (e.g., implementation details, ablation insights) we’ll explain step-by-step right away.
>
> Whether it’s about loss formulation, experimental validity, or other aspects please let us know! We’re committed to addressing every point thoroughly, even in these final hours. A quick reply (even a short question) would let us polish the paper.
>
> Thank you for your time, and we hope to hear from you soon.
>
> Best regards,
>
> CoIDO Team

---

> ### Comment · Reviewer_oB3E · 2025-08-09
> **Response to authors**
>
> Dear Authors,
>
> Thank you for your detailed rebuttal. I appreciate the time and effort you have put into addressing the points raised in my review. Your explanations have successfully resolved most of my concerns. I will raise my socre.

---

> ### Author Response · Authors · 2025-08-09
>
> Thank you sincerely for accepting our rebuttal and raising your score. We’re delighted that our responses have addressed your concerns. We're committed to refining our work further and look forward to continued engagement in advancing this research.

---

### Author Response · Authors · 2025-08-09
**Summary of Rebuttal & Discussion Phase**

We sincerely appreciate the efforts of AC, SAC, PC and all reviewers throughout the review process, as well as the valuable insights provided to help strengthen our work.

Reviewers have consistently recognized several key strengths of our submission. They noted that the core idea of Coupled Importance-Diversity Optimization (CoIDO) in the data selection domain is novel, with solid theoretical justifications. Additionally, they acknowledged that the proposed method achieves state-of-the-art performance and efficiency: compared to other data selection methods, CoIDO requires the **lowest computational resources (only one-third of others) and delivers the SOTA overall performance across ten benchmarks.**

A **common suggestion** from all reviewers was the need for further experiments on additional models and datasets to verify the method’s generalizability. In response, **we supplemented extensive experiments in our rebuttal, including evaluations on ``LLaVA-13B-LoRA``, ``LLaVA-150K``, and ``Vision-Flan.`` We also addressed each reviewer’s specific concerns in detail, providing targeted explanations and results.**

According to the feedback received, **all reviewers acknowledged that our supplementary experiments and explanations have resolved most of their concerns and expressed support for acceptance.**

---

### Decision · Program_Chairs · 2025-09-17

**Decision:**

Accept (poster)

**Comment:**

The final ratings for this paper are unanimously positive (two "Accept" and two "Boarderline Accept").

The AC agrees with the reviewers' positive evaluation. The paper introduces CoIDO framework for efficient data selection in visual instruction tuning. The primary concerns shared across all initial reviews are the method's generalizability and robustness.
However, the authors addressed these during the rebuttal by providing an set of additional experiments.
They demonstrated CoIDO's effectiveness on a larger model (LLaVA-13B-LoRA) and on two additional datasets (the smaller LLaVA-150K and the out-of-domain Vision-Flan). They also provided a detailed explanation for the Diversity Loss and presented new ablations on key hyperparameters.

The authors' rebuttal and the positive engagement from all reviewers have convincingly resolved the initial weaknesses. Therefore, the AC recommends this paper for acceptance.